# An Algebraic View of the Expressivity of Recurrent Language Models

**Franz Nowak** [1]  **Ryan Cotterell** [1]  **Reda Boumasmoud** [1]

## Abstract

What formal languages can a recurrent neural language model recognize? Formal results in the literature conflict: some authors report Turing-completeness, while others show equivalence to regular languages. The reason for this discrepancy is that the underlying arithmetic model differs. The paper develops a unified algebraic account of the expressivity of recurrent neural networks, starting with a formal account of various arithmetic models. This account reduces expressivity to an algebraic question, e.g., whether a network's syntactic monoid divides a certain wreath product. As a case study, the paper revisits diagonal state-space models: the same architecture cannot implement an even-modulus counter once floating-point recurrences are enforced, yet realizes every even-modulus counter under unsigned-integer quantization.

## 1. Introduction

The recent proliferation and widespread adoption of neural language models for a wide range of text-based tasks, e.g., question answering (OpenAI, 2021), proving mathematical statements (Lewkowycz et al., 2022), and general-purpose reasoning (Wei et al., 2022; OpenAI, 2024), have motivated a formal study of their capabilities. A growing research program treats architectures such as recurrent neural networks (RNNs) and transformer language models as objects of classical computation theory (Siegelmann and Sontag, 1992; Weiss et al., 2018; Merrill, 2019; Hahn, 2020; Bhattamishra et al., 2020; Pérez et al., 2021; Chiang and Cholak, 2022; Strobl et al., 2024, *inter alia*). They are studied as recognizers and generators of formal languages and classified by the computational resources they require.

[1]ETH Zürich. Correspondence to: Franz Nowak <franz.nowak@inf.ethz.ch>, Ryan Cotterell <ryan.cotterell@inf.ethz.ch>, Reda Boumasmoud <reda.boumasmoud@math.ethz.ch>.

*Proceedings of the $43^{rd}$ International Conference on Machine Learning*, Seoul, South Korea. PMLR 306, 2026. Copyright 2026 by the author(s).

This line of work on RNN expressivity has led to tangible practical insights: For instance, Sarrof et al. (2024) showed that linear RNNs with nonnegative diagonal recurrences, such as Mamba (Gu and Dao, 2024), cannot recognize languages that require maintaining a modulo counter over arbitrary lengths. This finding motivated the invention of more expressive linear RNNs (Grazzi et al., 2025).

Formal inquiry into the expressive power of neural language models, however, has produced seemingly contradictory conclusions. For example, it has been claimed that RNNs are Turing-complete (Siegelmann and Sontag, 1992). On the other hand, it has also been claimed that they can only simulate finite automata (Merrill, 2019). Both claims hold true, but, crucially, rely on different assumptions. For instance, some theoretical work assumes exact arithmetic over $\mathbb{R}$ or $\mathbb{Q}$ (Siegelmann and Sontag, 1992; Chen et al., 2018; Hahn, 2020; Bhattamishra et al., 2020; Svete et al., 2024), some treat precision as an explicit resource (e.g., logarithmic growth in sequence length) (Merrill and Sabharwal, 2023; Karuvally et al., 2025), while others appeal to finite precision while leaving operational details of the arithmetic underspecified (Omlin and Giles, 1996; Weiss et al., 2018; Sarrof et al., 2024; Stogin et al., 2024). To characterize an architecture's expressivity rigorously, these details must be fixed: the algebraic structure in which computation occurs, the representation of values, and the semantics of rounding, overflow, and evaluation order. Many proofs of expressivity implicitly rely on these identities and therefore do not automatically transfer to floating-point implementations. A central claim of this paper is that the underlying arithmetic semantics is part of the object being analyzed. It is an open question which assumptions are most appropriate for neural language models actually deployed in practice.

This paper brings together a variety of formal treatments of RNNs. Using concrete specifications of the underlying arithmetic model, we establish a purely algebraic characterization of recurrent architectures, unifying expressivity questions within an abstract framework. We show that arithmetic models with a finite set of possible values induce a finite, architecture-dependent transition monoid. In deep recurrent models, layers compose hierarchically, so multiple layers implement a cascade of algebraic cores. Algebraically, this cascade is a submonoid of the wreath product of the layer monoids. Expressivity questions then reduce to

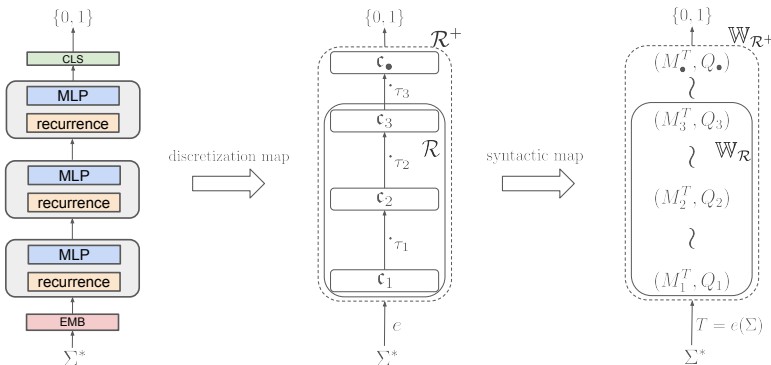

*Figure 1.* Abstraction of a deep RNN (left) to an algebraic RNN (center) to a wreath product of transformation monoids.

rigorous statements about the resulting monoid and its structure, recovering or refining many results in the literature as straightforward consequences. If any of the underlying assumptions change, such as the numerical semantics or the possible recurrence parametrization of a core, only this part needs to be adjusted while the remaining abstractions stay the same, greatly simplifying analysis.

We demonstrate the utility of our approach through a case study of diagonal SSMs, showing that different arithmetic semantics (numerical domain, rounding, evaluation order, etc.) give different expressivity results for the *same* idealized architecture. While nonnegative recurrence multipliers prevent diagonal SSMs from implementing counting in floating-point arithmetic, unsigned-integer quantization allows arbitrary even counters.

## 2. Algebraic Preliminaries

We collect the algebraic notions on which the rest of the paper rests. These notions are closely related to automata theory; the interested reader is referred to Pin (2010).

### 2.1. Alphabets, Words, and Languages

**Definition 2.1.** *An **alphabet** is a finite non-empty set $\Sigma$. The set $\Sigma^*$ of finite sequences (**words**) over $\Sigma$, equipped with concatenation and the empty word $\varepsilon$ as identity, is the **free monoid** on $\Sigma$.*

**Definition 2.2.** *A **language** over $\Sigma$ is a subset $\mathcal{L} \subseteq \Sigma^*$. Its **characteristic function** is*

$$\chi_\mathcal{L} \colon \Sigma^* \longrightarrow \{0, 1\}, \qquad \chi_\mathcal{L}(w) = \begin{cases} 1, & w \in \mathcal{L}, \\ 0, & w \notin \mathcal{L}. \end{cases}$$

We treat recurrent language models as *recognizers*: their role is to compute $\chi_\mathcal{L}$, not to define a distribution over $\Sigma^*$.

### 2.2. Monoids

**Definition 2.3.** *A **monoid** is a triple $(M, \cdot, 1_M)$ where $M$ is a set, $\cdot$ is an associative binary operation on $M$, and $1_M \in M$ satisfies $1_M \cdot m = m \cdot 1_M = m$ for all $m \in M$. We usually omit the operation and the identity from the notation and write $M$ for the monoid.*

The free monoid $\Sigma^*$ is the prototypical example: the identity is the empty word, and concatenation is associative.

**Definition 2.4.** *A **monoid morphism** between monoids $M$ and $N$ is a map $\phi \colon M \to N$ satisfying $\phi(1_M) = 1_N$ and $\phi(m_1 \cdot m_2) = \phi(m_1) \cdot \phi(m_2)$ for all $m_1, m_2 \in M$.*

A monoid morphism with domain $\Sigma^*$ is uniquely determined by its values on letters: any set map $\phi \colon \Sigma \to N$ extends uniquely to a monoid morphism $\widehat{\phi} \colon \Sigma^* \to N$.

### 2.3. Transformation Monoids

**Definition 2.5.** *For a set $Q$, the set $Q^Q$ of all functions $Q \to Q$ is a monoid under composition, with identity the identity function $\mathrm{id}_Q$. We call $Q^Q$ the **full transformation monoid** of $Q$.*

**Definition 2.6.** *A **transformation monoid** is a pair $(M, Q)$ consisting of a set $Q$ and a submonoid $M \leq Q^Q$. Equivalently, $M$ is a monoid equipped with a monoid morphism $\rho \colon M \to Q^Q$, called the **action**, with $\rho(1_M) = \mathrm{id}_Q$. We write $m \cdot q \overset{\text{def}}{=} \rho(m)(q)$ and identify $M$ with its image $\rho(M)$ when $\rho$ is injective.*

### 2.4. Composition of Transformation Monoids

We will repeatedly compose transformation monoids in two distinct ways: in *parallel*, modeling components that operate independently, and in *cascade*, modeling components arranged in a hierarchy where lower layers control the dynamics of upper ones.

**Definition 2.7.** *Let $(M_1, Q_1), \ldots, (M_n, Q_n)$ be transformation monoids. Their **parallel composition** (or direct prod-*

*uct) is the transformation monoid* $(M_1 \times \cdots \times M_n, Q_1 \times \cdots \times Q_n)$ *with componentwise action*

$$(m_1, \ldots, m_n) \cdot (q_1, \ldots, q_n) \overset{\text{def}}{=} (m_1 \cdot q_1, \ldots, m_n \cdot q_n).$$

*The identity is* $(\text{id}_{Q_1}, \ldots, \text{id}_{Q_n})$.

**Definition 2.8.** *Let* $(M_1, Q_1)$ *and* $(M_2, Q_2)$ *be transformation monoids. The **(left) wreath product** $(M_1, Q_1) \wr (M_2, Q_2)$ has underlying set $M_1 \times M_2^{Q_1}$ and state space $Q_1 \times Q_2$. Multiplication and action are*

$$(m_1, \phi_1)(m_2, \phi_2) \overset{\text{def}}{=} \big(m_1 m_2, q_1 \mapsto \phi_1(m_2 \cdot q_1)\phi_2(q_1)\big),$$
$$(m_1, \phi_1)(q_1, q_2) \overset{\text{def}}{=} \big(m_1 \cdot q_1, \phi_1(q_1) \cdot q_2\big),$$

*with identity* $(1_{M_1}, q_1 \mapsto 1_{M_2})$. *The wreath product of monoids is a monoid; of transformation monoids, a transformation monoid.*

The wreath product is the algebraic counterpart of cascade composition on the machine side: it captures the structure obtained when two transducers are composed in a hierarchy where the lower one controls the dynamics of the upper.

*Remark* 2.9 (Left convention). We adopt the *left* wreath product, where the upper-layer transition $\phi(q)$ depends on the *old* lower-layer state $q$. This matches the feedforward dependency in deep recurrent architectures, where the input delivered to layer $n+1$ at time $t$ is computed from the state of layer $n$ as it stands at the beginning of step $t$, before layer $n$ updates on the symbol at time $t$. The right wreath convention (more common in classical algebraic automata theory) is obtained by reversing operand order, but the algebraic content is the same.

*Remark* 2.10 (Associativity up to canonical isomorphism). The wreath product of transformation monoids is associative up to canonical isomorphism, i.e., the state spaces $(Q_1 \times Q_2) \times Q_3$ and $Q_1 \times (Q_2 \times Q_3)$ of the two bracketings of $(M_1, Q_1) \wr (M_2, Q_2) \wr (M_3, Q_3)$ differ, and the underlying monoids $(M_1 \times M_2^{Q_1}) \times M_3^{Q_1 \times Q_2}$ and $M_1 \times (M_2 \times M_3^{Q_2})^{Q_1}$ are identified by re-association and the Curry isomorphism $M_3^{Q_1 \times Q_2} \cong (M_3^{Q_2})^{Q_1}$. We henceforth write iterated wreath products without parentheses, equality between such expressions denoting the canonical isomorphism.

### 2.5. Division of Monoids

The right notion of "$M_1$ is structurally simpler than $M_2$" is not inclusion but *division*.

**Definition 2.11.** *A monoid $M_1$ **divides** a monoid $M_2$, written $M_1 \prec M_2$, if $M_1$ is a quotient of a submonoid of $M_2$: there exist a submonoid $N \leq M_2$ and a surjective monoid morphism $\pi \colon N \twoheadrightarrow M_1$. Division of transformation monoids is defined analogously, requiring $\pi$ to be compatible with the actions on the respective state spaces.*

Division induces a preorder on monoids. It is the relation that survives the abstraction from concrete dynamics to syntactic monoids: the syntactic monoid of any language recognized by a transducer divides its transition monoid, as recorded in Proposition 2.16.

### 2.6. Recognition of Languages by Monoids

The link between formal languages and monoids is given by Schützenberger's (1955) recognition theorem. Our presentation is based on Eilenberg (1976).

**Definition 2.12.** *A monoid $M$ **recognizes** a language $\mathcal{L} \subseteq \Sigma^*$ if there exist a monoid morphism $\eta \colon \Sigma^* \to M$ **and a** subset $P \subseteq M$ such that $\mathcal{L} = \eta^{-1}(P)$.*

**Theorem 2.13.** *For every language $\mathcal{L} \subseteq \Sigma^*$, there exists a unique-up-to-isomorphism monoid $M(\mathcal{L})$, the **syntactic monoid** of $\mathcal{L}$, characterized by the universal property that for any monoid $M$ and morphism $\eta \colon \Sigma^* \to M$ recognizing $\mathcal{L}$, the morphism $\eta$ factors through a surjective morphism onto $M(\mathcal{L})$. Equivalently, $M(\mathcal{L})$ is the quotient of $\Sigma^*$ by the **syntactic congruence***

$$u \sim_{\mathcal{L}} v \iff \forall x, y \in \Sigma^* \colon xuy \in \mathcal{L} \Leftrightarrow xvy \in \mathcal{L}.$$

*A language $\mathcal{L}$ is **regular** if and only if $M(\mathcal{L})$ is finite.*

The syntactic monoid is the smallest algebraic object that captures everything a monoid morphism out of $\Sigma^*$ can know about $\mathcal{L}$. Expressivity statements of the form *architecture $\mathcal{A}$ recognizes language $\mathcal{L}$* will, in this paper, factor through statements of the form $M(\mathcal{L})$ *divides the transition monoid of* $\mathcal{A}$.

### 2.7. Transducers and Acceptors

A useful intermediate object between a monoid morphism and a finite-state recognizer is the transducer: a state machine that processes a word symbol by symbol, updating an internal state and emitting an output at each step. We use it throughout the paper to model individual layers of a recurrent architecture.

**Definition 2.14.** *A **transducer** is a tuple $\mathcal{T} = (Q, q^\circ, X, Y, \delta, \gamma)$ where $Q$ is a state set with initial state $q^\circ \in Q$, $X$ is an input set, $Y$ is an output set, $\delta \colon Q \times X \to Q$ is the **transition map**, and $\gamma \colon Q \times X \to Y$ is the **output map**. Both maps are total.*

The transition map $\delta$ extends inductively to a map $\widehat{\delta} \colon Q \times X^* \to Q$ on words:

$$\widehat{\delta}(q, \varepsilon) = q, \qquad \widehat{\delta}(q, w \cdot x) = \delta(\widehat{\delta}(q, w), x).$$

The transducer above is in the *Mealy* form: the output $\gamma(q, x)$ depends on both the state and the current letter. When $Y = \{0, 1\}$, the transducer is an **acceptor**;

the language it recognizes consists of non-empty words $w = w' \cdot x \in X^+$, where $w' \in X^*$ and $x \in X$, satisfying $\gamma(\widehat{\delta}(q^\circ, w'), x) = 1$. Whether the empty word $\varepsilon$ is accepted requires an additional convention. We resolve this asymmetry in Section 3.6 by adopting a state-only readout for acceptance, where acceptance is a predicate on states and the empty word's status is determined by the initial state alone. For the Moore variant (output independent of input, $\gamma(q, x) = \gamma(q)$), the more familiar formula

$$\mathcal{L}(\mathcal{T}) = \{w \in X^* : \widehat{\delta}(q^\circ, w) \in F\}$$

with $F = \gamma^{-1}(1) \subseteq Q$ applies, and the empty word is accepted iff $q^\circ \in F$.

## 2.8. Transition Monoid of a Transducer

**Definition 2.15.** *Let $\mathcal{T}$ be a transducer. For each $x \in X$, the **induced transition** of $\mathcal{T}$ is*

$$\delta_x \colon Q \to Q, \qquad q \mapsto \delta(q, x).$$

*The **transition monoid** of $\mathcal{T}$ is*

$$T_{\mathcal{T}} \overset{\text{def}}{=} \big\langle \delta_x \mid x \in X \big\rangle_{\text{mon}} \le Q^Q,$$

*the submonoid of $Q^Q$ generated by the induced transitions, together with $\mathrm{id}_Q$.*

The transition monoid is the algebraic object that records all state transformations obtainable by feeding finite input words to $\mathcal{T}$, including the empty word.

**Proposition 2.16.** *If $\mathcal{T}$ is an acceptor recognizing $\mathcal{L} \subseteq X^*$, then the syntactic monoid $M(\mathcal{L})$ divides $T_{\mathcal{T}}$:*

$$M(\mathcal{L}) \prec T_{\mathcal{T}}.$$

This is the connection between the dynamical view (monoid of transformations of states) and the language view (syntactic monoid) that the rest of the paper exploits.

# 3. Algebraic RNNs

Recurrent language models, in their analytic form, are typically defined in terms of real-valued tensors, smooth nonlinearities, and parameter matrices. The paper strips this down to the structure that *survives* discretization: a typed family of total maps and their iterated composition, specified combinatorially before any choice of arithmetic semantics. The arithmetic then enters as a parameter, fixing the concrete sets and transition functions each algebraic core realizes.

## 3.1. Algebraic Cores

**Definition 3.1.** *An **algebraic core** is a tuple $\mathfrak{c} = (Q, X, Y, f, g)$ where $Q$, $X$, and $Y$ are sets, and*

$$f \colon Q \times X \to Q, \qquad g \colon Q \times X \to Y$$

*are total maps, called respectively the **transition** and **readout** functions.*

**Mealy, not Moore.** When $Q, X, Y$ are finite, $\mathfrak{c}$ is precisely a Mealy machine (Mealy, 1955), the standard finite-state transducer of classical automata theory (Eilenberg, 1976). The fact that $g$ depends on *both* the state *and* the current letter–not on the state alone–distinguishes Mealy machines from their Moore counterparts (Moore, 1956). We retain this Mealy-style readout (depending on both state and input) because the recurrent layers we wish to abstract (e.g., Elman, SSM) all share it.

**Definition 3.2.** *Let $\mathfrak{c}$ be an algebraic core. For each $x \in X$, the **induced transition** is*

$$f_x \colon Q \to Q, \qquad q \mapsto f(q, x).$$

*The **transition monoid** of $\mathfrak{c}$ is the submonoid of $Q^Q$ generated by these maps,*

$$M_{\mathfrak{c}} \overset{\text{def}}{=} \big\langle f_x \mid x \in X \big\rangle_{\text{mon}} \le Q^Q,$$

*and the pair $(M_{\mathfrak{c}}, Q)$ is the associated **transformation monoid**.*

*Remark* 3.3 (Dynamics versus observability). The readout $g$ contributes nothing to $M_{\mathfrak{c}}$. The monoid encodes the *dynamics* reachable on $Q$; the readout encodes *observability*. This separation will matter: the same dynamical core can support distinct readouts, and a given language is recognized by a *quotient* of the dynamics determined by its observable consequences. We return to this point in Section 3.6.

## 3.2. Algebraic RNNs

**Definition 3.4.** *An **algebraic RNN** of depth $N$ is a tuple*

$$\mathcal{R} = \big(\mathfrak{c}_1, (\mathfrak{c}_2, \tau_1), \ldots, (\mathfrak{c}_N, \tau_{N-1})\big),$$

*where each $\mathfrak{c}_n = (Q_n, X_n, Y_n, f_n, g_n)$ is an algebraic core, and each $\tau_{n-1} \colon X_{n-1} \times Y_{n-1} \to X_n$ for $2 \le n \le N$ is a **wiring map** from $\mathfrak{c}_{n-1}$ to $\mathfrak{c}_n$. The first core has no incoming wiring. The **global state space** of $\mathcal{R}$ is*

$$Q_{\mathcal{R}} \overset{\text{def}}{=} Q_1 \times \cdots \times Q_N.$$

*The **input type** of $\mathcal{R}$ is the set $\mathrm{In}(\mathcal{R}) \overset{\text{def}}{=} X_1$, and the **output type** of $\mathcal{R}$ is the pair $\mathrm{Out}(\mathcal{R}) \overset{\text{def}}{=} (X_N, Y_N)$.*

The asymmetry between the two reflects the type-signature of wiring maps, which consume a pair $X \times Y$ to produce an input. The pair $(\mathrm{In}(\mathcal{R}), \mathrm{Out}(\mathcal{R}))$ records precisely the external interface of $\mathcal{R}$.

An algebraic RNN is, on its own, neither an acceptor nor a generator: it is best viewed as a transducer. The two further pieces of structure needed to turn it into a language acceptor (an initial state, an encoder, an accepting core) are deferred to Section 3.6. Before adding them, we identify the algebraic object inside which $\mathcal{R}$ already lives.

## 3.3. Cascade Juxtaposition

The construction of an algebraic RNN by appending a new core to a cascade, as an accepting core, an initialization core, or a post-processing core, appears repeatedly throughout this paper. We give the operation a formal name and observe that, with input and output types separated, it becomes the associative composition of a typed category.

**Wirings as type morphisms.** Given an algebraic RNN $\mathcal{R}$ with output type $\mathrm{Out}(\mathcal{R}) = (X_N, Y_N)$ and an algebraic RNN $\mathcal{R}'$ with input type $\mathrm{In}(\mathcal{R}') = X_1'$, a **wiring map** from $\mathcal{R}$ to $\mathcal{R}'$ is a function

$$\tau \colon \mathrm{Out}(\mathcal{R}) \;\longrightarrow\; \mathrm{In}(\mathcal{R}'),$$

i.e., a map $X_N \times Y_N \to X_1'$. The output type of $\mathcal{R}$ and the input type of $\mathcal{R}'$ thus carry all the information needed to specify $\tau$'s domain and codomain.

**Definition 3.5.** *Let $\mathcal{R}$ and $\mathcal{R}'$ be algebraic RNNs of depths $N$ and $N'$, and let $\tau \colon \mathrm{Out}(\mathcal{R}) \to \mathrm{In}(\mathcal{R}')$ be a wiring map between them. The **cascade juxtaposition** of $\mathcal{R}$ and $\mathcal{R}'$ along $\tau$ is the algebraic RNN of depth $N + N'$ obtained by concatenating their sequences of pairs:*

$$\mathcal{R} \rhd_\tau \mathcal{R}' \stackrel{\text{def}}{=} (\mathfrak{c}_1, \ldots, (\mathfrak{c}_N, \tau_{N-1}), (\mathfrak{c}_1', \tau), \ldots, (\mathfrak{c}_{N'}', \tau_{N'-1}'))$$

*where the wiring of the first core of $\mathcal{R}'$ is taken to be $\tau$. The juxtaposition has input type $\mathrm{In}(\mathcal{R})$ and output type $\mathrm{Out}(\mathcal{R}')$:*

$$\mathrm{In}(\mathcal{R} \rhd_\tau \mathcal{R}') = \mathrm{In}(\mathcal{R}), \qquad \mathrm{Out}(\mathcal{R} \rhd_\tau \mathcal{R}') = \mathrm{Out}(\mathcal{R}').$$

*The special case where $\mathcal{R}'$ is the depth-one RNN with a single pair $(\mathfrak{c}', \tau)$ recovers the operation of appending a single core to a cascade.*

*Remark* 3.6. Cascade juxtaposition is associative, once input and output types are tracked. Given three algebraic RNNs $\mathcal{R}_1, \mathcal{R}_2, \mathcal{R}_3$ and wiring maps $\tau_{12} \colon \mathrm{Out}(\mathcal{R}_1) \to \mathrm{In}(\mathcal{R}_2)$, $\tau_{23} \colon \mathrm{Out}(\mathcal{R}_2) \to \mathrm{In}(\mathcal{R}_3)$, the type of $\tau_{23}$ depends only on $\mathrm{Out}(\mathcal{R}_2)$ and $\mathrm{In}(\mathcal{R}_3)$, both unchanged by either parenthesization. The two expressions $(\mathcal{R}_1 \rhd_{\tau_{12}} \mathcal{R}_2) \rhd_{\tau_{23}} \mathcal{R}_3$ and $\mathcal{R}_1 \rhd_{\tau_{12}} (\mathcal{R}_2 \rhd_{\tau_{23}} \mathcal{R}_3)$ produce the same sequence of pairs $(\mathfrak{c}_n, \tau_{n-1})$ and hence, by Definition 3.4, denote the same algebraic RNN. Iterated juxtaposition $\mathfrak{c}_1 \rhd_{\tau_2} \mathfrak{c}_2 \rhd_{\tau_3} \cdots \rhd_{\tau_N} \mathfrak{c}_N$ is therefore well-defined once the typed sequence of cores and wirings is fixed.

*Remark* 3.7 (General wirings). The wiring maps of Definition 3.4 are *local*: each $\tau_n$ depends only on the input and output of the immediately preceding core $\mathfrak{c}_{n-1}$. This convention is sufficient to model the standard recurrent architectures.

The framework extends, however, to *general wirings*

$$\tau_n \colon X_{\leq n-1} \times Y_{\leq n-1} \to X_n,$$

where $X_{\leq n-1} \stackrel{\text{def}}{=} X_1 \times \cdots \times X_{n-1}$ and similarly for $Y_{\leq n-1}$. Such wirings give layer $n$ direct access to the inputs and outputs of *all* preceding layers, modeling arbitrary skip connections, dense connections, and residual shortcuts that bypass intermediate layers.

The transition to general wirings affects only one definition in the body of the paper. The layer-input dependency map (Definition 3.9) acquires the corresponding extended recursion: writing $x_n = \varphi_n^T(q_{<n}, t)$ and $y_n = g_n(f_n(q_n, x_n), x_n)$,

$$\varphi_{n+1}^T(q_{<n+1}, t) \stackrel{\text{def}}{=} \tau_n\big((x_1, \ldots, x_n), (y_1, \ldots, y_n)\big).$$

Every other algebraic statement of the paper passes through verbatim. The factorization lemma (Lemma 3.12) still holds: the induction on depth uses only that the input to layer $n+1$ is determined by $(q_{<n+1}, t)$, which the general wiring respects. The cascade juxtaposition identity (Proposition 3.20) remains valid: the projection $\pi_{\leq N}$ is still surjective onto the dynamics of the lower segment. The syntactic divisibility chain leading to $M(\mathcal{L}(\mathcal{A})) \prec W_{\mathcal{R}^+}^{e(\Sigma)}$ holds as before.

The price of generality is notational. Under general wirings, the output type of $\mathcal{R}$ inflates from the pair $(X_N, Y_N)$ to the full history $(X_{\leq N}, Y_{\leq N})$. Cascade juxtaposition $\mathcal{R} \rhd_\tau \mathcal{R}'$ then takes a wiring $\tau \colon X_{\leq N} \times Y_{\leq N} \to X_1'$, and the typed associativity of Remark 3.6 holds modulo this enriched type accounting.

We adopt the local form throughout for notational simplicity, with the understanding that none of the algebraic results of the paper are specific to it.

## 3.4. The Ambient Wreath Product

Write $M_n \stackrel{\text{def}}{=} M_{\mathfrak{c}_n}$ for the transition monoid of the $n^{\text{th}}$ core, and $(M_n, Q_n)$ for its transformation monoid.

**Definition 3.8.** *The **ambient wreath product** of $\mathcal{R}$ is the iterated transformation monoid*

$$\mathbb{W}_1 \stackrel{\text{def}}{=} (M_1, Q_1), \qquad \mathbb{W}_{n+1} \stackrel{\text{def}}{=} \mathbb{W}_n \wr (M_{n+1}, Q_{n+1}).$$

*We write $\mathbb{W}_{\mathcal{R}} \stackrel{\text{def}}{=} \mathbb{W}_N = (W_{\mathcal{R}}, Q_{\mathcal{R}})$. By construction, $(W_{\mathcal{R}}, Q_{\mathcal{R}}) \leq (Q_{\mathcal{R}}^{Q_{\mathcal{R}}}, Q_{\mathcal{R}})$ is a transformation monoid on the global state space.*

The ambient wreath product of $\mathcal{R}$ captures all cores and their hierarchical dependency, but is blind to the wiring maps $\tau_n$. Every layer $n$ may, in $\mathbb{W}_{\mathcal{R}}$, be driven by any input in $X_n$, regardless of which inputs the wiring actually activates. The realized dynamics live in a strictly smaller sub-object.

## 3.5. Realized Dynamics: a Parameterized Construction

We now refine $\mathbb{W}_{\mathcal{R}}$ to a sub-transformation-monoid that respects the wiring. We parameterize the construction by an

arbitrary set $T \subseteq X_1$ of admissible inputs to the first core, prove a single factorization lemma at this level of generality, and recover the two cases of interest (the unconstrained RNN and the RNN as language acceptor) as specializations.

**Definition 3.9.** *Let $\mathcal{R}$ be an algebraic RNN of depth $N$ and let $T \subseteq X_1$. The **layer-input dependency maps** of $\mathcal{R}$ relative to $T$, denoted $\varphi_n^T$, are defined as follows*

$$\varphi_n^T \colon Q_{<n} \times T \to X_n, \quad Q_{<n} \overset{\text{def}}{=} Q_1 \times \cdots \times Q_{n-1}$$

*(with $Q_{<1}$ a singleton, so that $\varphi_1^T$ is essentially the map $T \to X_1, t \mapsto t$) are defined inductively: For $1 < n \le N$, writing $x_n = \varphi_n^T(q_{<n}, t)$,*

$$\varphi_{n+1}^T(q_{<n+1}, t) \overset{\text{def}}{=} \tau_{n-1}\Big(x_n, g_n\big(f_n(q_n, x_n), x_n\big)\Big).$$

*The **reachable input set**[1] at layer $n$ is $X_n^T \overset{\text{def}}{=} \operatorname{Im}(\varphi_n^T) \subseteq X_n$.*

The set $X_n^T$ is the set of inputs that the wiring can actually deliver to layer $n$ when the first core is driven by letters from $T$.

**Definition 3.10.** *The **realized transition monoid** of layer $n$ relative to $T$ is*

$$M_n^T \overset{\text{def}}{=} \big\langle (f_n)_x \mid x \in X_n^T \big\rangle_{\text{mon}} \le M_n.$$

**Definition 3.11.** *The **realized wreath product** of $\mathcal{R}$ relative to $T$ is*

$$\mathbb{W}_{\mathcal{R}}^T \overset{\text{def}}{=} (M_1^T, Q_1) \wr \cdots \wr (M_N^T, Q_N) \le \mathbb{W}_{\mathcal{R}},$$

*with underlying monoid $W_{\mathcal{R}}^T$.*

**Lemma 3.12** (Factorization). *Let $T \subseteq X_1$. For each $t \in T$, the global state update*

$$(F_{\mathcal{R}})_t \colon Q_{\mathcal{R}} \to Q_{\mathcal{R}}$$

*induced by feeding $t$ into the first core is an element of $W_{\mathcal{R}}^T$. Consequently, the global transition monoid driven by $T$,*

$$M_{\mathcal{R}}^T \overset{\text{def}}{=} \big\langle (F_{\mathcal{R}})_t \mid t \in T \big\rangle_{\text{mon}},$$

*satisfies*

$$M_{\mathcal{R}}^T \le W_{\mathcal{R}}^T \le W_{\mathcal{R}}.$$

*Proof sketch.* Induction on $N$, as in Appendix E. The base case is immediate. For the inductive step, the global update splits as $(F_{\mathcal{R}})_t(q, q_N) = (\bar{F}(q), \psi_{\bar{F}, t}(q)(q_N))$ for $q \in Q_{<N+1}$, where $\bar{F} \in W_{\mathcal{R}|_{<N}}^T$ by induction and $\psi_{\bar{F}, t}(q) = (f_N)_{\varphi_N^T(q,t)} \in M_N^T$ by Definition 3.10. This is precisely the action of an element of the wreath product. The action of the empty word in $T^*$ corresponds to $\operatorname{id}_{Q_{\mathcal{R}}} \in W_{\mathcal{R}}^T$. $\square$

---

[1]Sometimes called the *effective* or *realized* input set in the literature; we use reachable to emphasize that this set is the image of the layer-input dependency map starting from $T$.

*Remark* 3.13 (Functoriality in $T$). The assignment $T \mapsto \mathbb{W}_{\mathcal{R}}^T$ is a covariant functor from the poset $(\mathscr{P}(X_1), \subseteq)$ to the poset of transformation submonoids of $\mathbb{W}_{\mathcal{R}}$, ordered by inclusion (see Remark E.1). Explicitly: if $T \subseteq T'$, then $X_n^T \subseteq X_n^{T'}$ at every layer, hence $M_n^T \le M_n^{T'}$ and $\mathbb{W}_{\mathcal{R}}^T \le \mathbb{W}_{\mathcal{R}}^{T'}$. The ambient case is the terminal value: $\mathbb{W}_{\mathcal{R}}^{X_1} = \mathbb{W}_{\mathcal{R}}$.

Two specializations of Lemma 3.12 are of primary interest. The terminal case $T = X_1$ recovers the ambient wreath product ($\mathbb{W}_{\mathcal{R}}^{X_1} = \mathbb{W}_{\mathcal{R}}$, by Remark 3.13) and is the relevant setting when $\mathcal{R}$ is viewed as an autonomous transducer freely driven by inputs to its first core. The case $T = e(\Sigma)$, for some encoder $e \colon \Sigma \to X_1$ of a finite alphabet $\Sigma$, is the relevant setting when $\mathcal{R}$ is the dynamical backbone of a language acceptor (Section 3.6): the realized wreath product $\mathbb{W}_{\mathcal{R}}^{e(\Sigma)}$ then governs the dynamics reachable by feeding $\Sigma^*$ into $\mathcal{R}$, and divides the recognized language's algebra.

### 3.6. Algebraic RNNs as Language Acceptors

#### 3.6.1. ACCEPTING CORE

A classical automaton accepts via a final-state set $F \subseteq Q$. The algebraic RNNs of Definition 3.4 carry instead, in their last core, a Mealy-style readout $g_N \colon Q_N \times X_N \to Y_N$ whose value depends jointly on the post-update state *and* the last input read. Composing with a decoder $d \colon Y_N \to \{0, 1\}$ thus yields not a subset of $Q_N$ but a subset of $Q_N \times X_N$. The acceptance condition becomes a predicate on *(state, last symbol)* pairs rather than on states alone, and the algebraic recognition framework of Eilenberg (1976) does not apply directly.

We resolve this not by external bookkeeping, as would be done by reserving a dedicated end-of-sequence symbol, but *structurally*, by absorbing the decoder into the algebra itself. The decoder $d$ is realized as an additional algebraic core, appended to the architecture, whose internal state evolves under the same wiring discipline as every other layer and whose role is to contain the acceptance condition as a property of its own state.

**Definition 3.14.** *An **accepting core** is an algebraic core $\mathfrak{c}_\bullet = (Q_\bullet, X_\bullet, \{0, 1\}, f_\bullet, g_\bullet)$ whose readout depends only on the state: there exists a subset $F \subseteq Q_\bullet$, called the **accepting region**, such that*

$$g_\bullet(q, x) = \chi_F(q) \quad \text{for all } q \in Q_\bullet, \ x \in X_\bullet,$$

*where $\chi_F \colon Q_\bullet \to \{0, 1\}$ is the characteristic function of $F$.*

*Remark* 3.15 (Moore content). Definition 3.14 captures, in algebraic terms, exactly the structure of a Moore machine with Boolean output: the readout depends only on the state, which is what turns acceptance into a predicate on states rather than a predicate on state-letter pairs. When $Q_\bullet$ is a singleton, $\mathfrak{c}_\bullet$ degenerates to a pure set map $X_\bullet \to \{0, 1\}$, recovering the memoryless decoder $d \colon Y_N \to \{0, 1\}$ of the

literature (precomposed with the wiring map). The present formulation thus strictly generalizes that one: it allows the decoder to be itself stateful, while specializing back to a memoryless decision in the singleton case.

### 3.6.2. RNN ACCEPTORS

**Definition 3.16.** *An **RNN acceptor** is a tuple $\mathscr{A} = (\Sigma, \mathscr{R}^+, q^\circ, e)$ where where $\mathscr{R}^+ = \mathscr{R} \rhd_{\tau_N} \mathfrak{c}_\bullet$ is the augmented RNN obtained by appending an accepting core $\mathfrak{c}_\bullet$ to a base RNN $\mathscr{R}$ via a wiring map $\tau_N$, and:*

- *$\Sigma$ is a finite alphabet;*
- *$\mathscr{R}$ is an algebraic RNN of depth $N$, which we name the **base RNN** of $\mathscr{A}$;*
- *$\mathfrak{c}_\bullet$ is an accepting core (Definition 3.14);*
- *$\tau_N \colon X_N \times Y_N \to X_\bullet$ is the wiring map between $\mathscr{R}$ and $\mathfrak{c}_\bullet$;*
- *$q^\circ \in Q_{\mathscr{R}+}$ is an initial global state;*
- *$e \colon \Sigma \to X_1$ is the encoder.*

*The wiring map is, in every case, part of the acceptor's specification.*

*Remark* 3.17 (The role of $\mathscr{R}^+$). The augmented RNN $\mathscr{R}^+$ is, by Definition 3.4, a genuine algebraic RNN: the Moore constraint on $\mathfrak{c}_\bullet$ acts only at the last layer and does not affect the application of the algebraic theory of Section 3.5 to $\mathscr{R}^+$.

*Remark* 3.18 (The acceptor as a sandwich). The definition of an RNN acceptor (Definition 3.16) treats $q^\circ$ and $F$ asymmetrically: the initial state $q^\circ$ enters as direct primitive data, while the accepting region $F$ is wrapped inside the accepting core $\mathfrak{c}_\bullet$ as part of its algebraic structure. This asymmetry is dispensable. Indeed, the initial state, like the accepting core, can be internalized into an algebraic core as well, placed at the input end of the cascade. Concretely, an **initialization core** is an algebraic core $\mathfrak{c}_\circ = (Q_\circ, X_\circ, Y_\circ, f_\circ, g_\circ)$ equipped with a distinguished input $\sigma_* \in X_\circ$ and a base state $q_\circ^0 \in Q_\circ$, such that $f_\circ(q_\circ^0, \sigma_*)$ delivers, via an outgoing wiring map, the desired initial state $q^\circ$ to the next core, and such that $f_\circ(q, x) = q$ for all $q \neq q_\circ^0$ and all $x \in X_\circ$. The **dualized acceptor**

$$\mathscr{A}^\circ = \big(\Sigma \cup \{\sigma_*\}, \mathfrak{c}_\circ \rhd_{\tau_0} \mathscr{R} \rhd_{\tau_N} \mathfrak{c}_\bullet, q_\circ^0, e^\circ\big),$$

where $\sigma_* \notin \Sigma$ is a distinguished prefix symbol, is constructed by juxtaposing $\mathfrak{c}_\circ$ to the left of $\mathscr{R}$ via $\tau_0$, and reading $\sigma_* \cdot w$ in place of $w$. It recognizes exactly $\mathcal{L}(\mathscr{A})$, with identical syntactic monoid $M(\mathcal{L}(\mathscr{A}^\circ)) = M(\mathcal{L}(\mathscr{A}))$. A general acceptor is thus, in its most symmetric form, a *sandwich*

$$\mathfrak{c}_\circ \rhd_{\tau_0} \mathscr{R} \rhd_{\tau_N} \mathfrak{c}_\bullet,$$

with an initialization core $\mathfrak{c}_\circ$ at the input end of the cascade and an accepting core $\mathfrak{c}_\bullet$ at the output end. The initialization

core contributes a one-element submonoid to the realized wreath product (the identity, after its first action), so divisibility arguments collapse its contribution to triviality: the dualization is harmless at the level of recognized languages and their syntactic monoids. We do not adopt the dualized form as a working definition. The standard formulation (Definition 3.16) is closer to the machine-learning idiom in which an RNN's initial state is part of its specification.

### 3.6.3. ALGEBRAIC CONTENT

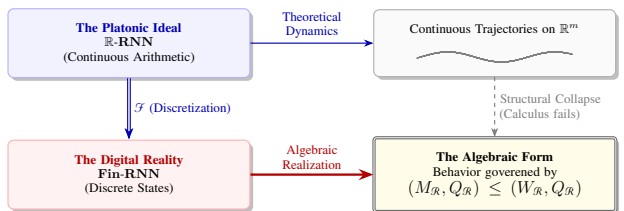

*Figure 2.* A structural perspective: from continuous idealizations (top) to algebraic structures (bottom).

**Definition 3.19.** *Let $\mathscr{A} = (\Sigma, \mathscr{R}^+, q^\circ, e)$ be an RNN acceptor. For $w = \sigma_1 \cdots \sigma_n \in \Sigma^*$, write $\widehat{F}_{\mathscr{R}+}(q^\circ, w) \in Q_{\mathscr{R}+}$ for the global state obtained by iterating $\mathscr{R}^+$ from $q^\circ$ on the encoded sequence $e(\sigma_1), \ldots, e(\sigma_n)$, with the convention that the empty word acts as $\mathrm{id}_{Q_{\mathscr{R}+}}$, so that in particular $\widehat{F}_{\mathscr{R}+}(q^\circ, \varepsilon) = q^\circ$. The **language recognized by** $\mathscr{A}$ is*

$$\mathcal{L}(\mathscr{A}) \overset{\text{def}}{=} \Big\{ w \in \Sigma^* \;\Big|\; \pi_{N+1}\big(\widehat{F}_{\mathscr{R}+}(q^\circ, w)\big) \in F \Big\},$$

*where $\pi_{N+1} \colon Q_{\mathscr{R}+} \to Q_{N+1}$ is the projection onto the accepting core's state coordinate.*

The dynamics relevant for $\mathcal{L}(\mathscr{A})$ live in the realized wreath product $\mathbb{W}_{\mathscr{R}+}^{e(\Sigma)}$ associated with the augmented network and the admissible-input set $T = e(\Sigma) \subseteq X_1$. The encoder lifts to a monoid morphism

$$\Phi_{\mathscr{A}} \colon \Sigma^* \longrightarrow M_{\mathscr{R}+}^{e(\Sigma)}, \quad w \mapsto \widehat{F}_{\mathscr{R}+}(-, w),$$

and

$$\mathcal{L}(\mathscr{A}) = \Phi_{\mathscr{A}}^{-1}(P_F),$$

where $P_F \overset{\text{def}}{=} \{m \in M_{\mathscr{R}+}^{e(\Sigma)} : \pi_{N+1}(m \cdot q^\circ) \in F\}$. The syntactic monoid $M(\mathcal{L}(\mathscr{A}))$ therefore divides $M_{\mathscr{R}+}^{e(\Sigma)}$, and by Lemma 3.12 divides the realized wreath product $W_{\mathscr{R}+}^{e(\Sigma)}$, which in turn divides the ambient wreath product $W_{\mathscr{R}+}$:

$$M(\mathcal{L}(\mathscr{A})) \prec M_{\mathscr{R}+}^{e(\Sigma)} \leq W_{\mathscr{R}+}^{e(\Sigma)} \leq W_{\mathscr{R}+}. \tag{3.1}$$

**Decomposition of the recognition problem.** The divisibility chain above turns the recognition question into a question about finite monoids. Fix a family of algebraic RNNs, a choice of permissible cores, wirings, and arithmetic semantics fixed once and for all, and a target language

$\mathcal{L} \subseteq \Sigma^*$. The question *does some acceptor in the family recognize $\mathcal{L}$?* decomposes as follows:

(i) *Algebraic impossibility.* The syntactic monoid $M(\mathcal{L})$ must divide some realized wreath product $W_{\mathcal{R}^+}^{e(\Sigma)}$ achievable in the family. Krohn–Rhodes theory (Appendix I) decomposes $M(\mathcal{L})$ into flip-flops and prime transformation groups. If any required factor cannot be realized as a layer monoid permitted in the family, no acceptor in the family recognizes $\mathcal{L}$.

(ii) *Observational realization.* Conversely, to exhibit a recognizer, one constructs cores realizing the Krohn–Rhodes factors of $M(\mathcal{L})$, wires them so that the realized wreath product contains $M(\mathcal{L})$ as a divisor, and chooses the accepting region $F$ to separate the syntactic congruence classes through the readout-induced quotient.

This decomposition applies uniformly to any architecture family fitting Definition 3.4 under any arithmetic semantics fixed as in Section 4. The case study of Appendix J carries out both directions on diagonal SSMs: the algebraic side rules out modular counting under floating-point recurrences, while the observational side constructs an explicit acceptor for parity under unsigned-integer quantization.

### 3.6.4. THE AUGMENTATION AS A WREATH EXTENSION

The cascade juxtaposition $\mathcal{R} \rhd_{\tau_N} \mathfrak{c}_\bullet$ of Definition 3.16 takes a base RNN $\mathcal{R}$ and an accepting core $\mathfrak{c}_\bullet$, glues them by a wiring map $\tau_N$, and produces a deeper algebraic RNN. At the level of realized dynamics, the cascade juxtaposition becomes a wreath product: the realized monoid of the augmented RNN sits inside the wreath product of the realized monoid of $\mathcal{R}$ and the realized monoid of $\mathfrak{c}_\bullet$. This sharpens Lemma 3.12 for the acceptor case and isolates the algebraic role of the accepting core.

**Proposition 3.20.** *Let $\mathcal{R}$ and $\mathcal{R}'$ be algebraic RNNs of depths $N$ and $N'$ respectively, and let $\tau \colon \mathrm{Out}(\mathcal{R}) \to \mathrm{In}(\mathcal{R}')$ be a wiring map between them. For any $T \subseteq \mathrm{In}(\mathcal{R})$,*

$$\mathbb{W}_{\mathcal{R} \rhd_\tau \mathcal{R}'}^T = \mathbb{W}_{\mathcal{R}}^T \wr \mathbb{W}_{\mathcal{R}'}^T,$$

*where the equality is understood up to canonical isomorphism (Remark 2.10), and $\mathbb{W}_{\mathcal{R}'}^T$ denotes the realized wreath product of $\mathcal{R}'$ relative to the set of inputs reachable through $\tau$ from states of $\mathcal{R}$ activated by $T$. The realized transition monoid of the juxtaposition satisfies the chain*

$$M_{\mathcal{R}}^T \prec M_{\mathcal{R} \rhd_\tau \mathcal{R}'}^T \leq W_{\mathcal{R}}^T \wr W_{\mathcal{R}'}^T,$$

*the first division being given by the canonical projection $\pi_{\leq N} \colon Q_{\mathcal{R} \rhd_\tau \mathcal{R}'} \to Q_{\mathcal{R}}$.*

*Proof.* See Appendix F. □

**Corollary 3.21** (Augmentation of an acceptor). *Let $\mathcal{A} = \left(\Sigma, \mathcal{R}^+, q^\circ, e\right)$ be an RNN acceptor with base RNN $\mathcal{R}$ of depth $N$, accepting core $\mathfrak{c}_\bullet$ and wiring map $\tau_N$, and let $T = e(\Sigma) \subseteq X_1$. Then*

$$W_{\mathcal{R}^+}^T = W_{\mathcal{R}}^T \wr M_{\mathfrak{c}_\bullet}^T,$$

*again up to canonical isomorphism, and the chain*

$$M_{\mathcal{R}}^T \prec M_{\mathcal{R}^+}^T \leq W_{\mathcal{R}}^T \wr M_{\mathfrak{c}_\bullet}^T$$

*holds, with $M_{\mathfrak{c}_\bullet}^T \leq M_{\mathfrak{c}_\bullet}$ the realized transition monoid of $\mathfrak{c}_\bullet$ relative to the inputs it receives through $\tau_N$.*

*Proof.* Specialization of Proposition 3.20 to the case where $\mathcal{R}'$ is the depth-one RNN consisting of $\mathfrak{c}_\bullet$ alone; the realized wreath product $W_{\mathcal{R}'}^T$ then reduces to $M_{\mathfrak{c}_\bullet}^T$. □

*Remark* 3.22 (The accepting core sits above the dynamics). Corollary 3.21 makes explicit the following: the augmentation by an accepting core *adds expressive resolution above the base RNN without modifying the dynamics on $Q_{\mathcal{R}}$.* For divisibility-based expressivity arguments, the consequence is: $M(\mathcal{L}(\mathcal{A}))$ divides the augmented monoid $M_{\mathcal{R} \rhd_{\tau_N} \mathfrak{c}_\bullet}^T$, but not necessarily the base monoid $M_{\mathcal{R}}^T$. The accepting core may, in general, extract recognition power that the base alone cannot encode, exactly when its own state space $Q_\bullet$ has cardinality greater than one. The interest of the accepting-core formulation, compared to a memoryless decoder, depends on the cardinality of $Q_\bullet$:

(i) If $|Q_\bullet| = 1$, then $M_{\mathfrak{c}_\bullet} = \{\mathrm{id}_{Q_\bullet}\}$ and the wreath extension collapses, $W_{\mathcal{R}}^T \wr M_{\mathfrak{c}_\bullet}^T \cong W_{\mathcal{R}}^T$. The acceptor reduces to a pair $(\mathcal{R}, F \subseteq Y_N)$ where $F$ is a Boolean subset of the base RNN's last readout values. This yields the formulation of a recognizer found in the literature.

(ii) When $|Q_\bullet| > 1$, the accepting core carries *stateful memory* above the base dynamics and may extract regular languages that the memoryless case cannot reach.

## 4. The Impact of Discretization

### 4.1. Algebraic Properties of Finite-Precision Semantics

Digital computers store numbers in discrete, finite data types. They therefore cannot represent arbitrary real values. Instead, computation is carried out over a finite representable set FP $\subseteq \mathbb{R}$, and the result of each arithmetic operation is mapped back into FP by rounding or truncation.

For formal expressivity analysis, reasoning over $\mathbb{R}$ while treating finite precision as a small approximation error fails: quantization alters the very algebraic structure on which expressivity statements rest.

**Definition 4.1** (Evaluation tree). *An **evaluation tree** over a set of binary operations $\mathcal{O}$ in $k$ variables is a finite rooted ordered binary tree whose leaves are labeled by variables $x_1, \ldots, x_k$ or by constants, and whose internal nodes are labeled by operations in $\mathcal{O}$. A tree thus encodes a parenthesized expression.*

**Definition 4.2.** *An **arithmetic model** is a triple $\mathfrak{M} = (\mathcal{D}, \mathcal{O}, \square)$ with $\mathcal{D}$ a value domain in an ambient set (usually $\mathbb{R}$), $\mathcal{O}$ a finite set of binary operations on $\mathcal{D}$, and $\square$ a rounding map from the ambient set into $\mathcal{D}$. For an evaluation tree $\mathcal{T}$ over $\mathcal{O}$ with constants in $\mathcal{D}$, the **evaluation under** $\mathfrak{M}$ is the function $[\![\mathcal{T}]\!]_{\mathfrak{M}} : \mathcal{D}^k \to \mathcal{D}$ defined by induction: at a variable leaf $x_j$, $[\![\mathcal{T}]\!]_{\mathfrak{M}} \overset{\text{def}}{=} \pi_j$ (the $j$-th projection); at a constant leaf $c$, $[\![\mathcal{T}]\!]_{\mathfrak{M}} \overset{\text{def}}{=} c$; at an internal node $\mathcal{T} = \mathcal{T}_1 \odot \mathcal{T}_2$, $[\![\mathcal{T}]\!]_{\mathfrak{M}} \overset{\text{def}}{=} \square([\![\mathcal{T}_1]\!]_{\mathfrak{M}} \odot [\![\mathcal{T}_2]\!]_{\mathfrak{M}})$. An expression $f$ admits, in general, several evaluation trees; the abbreviation $[\![f]\!]_{\mathfrak{M}} \overset{\text{def}}{=} [\![\mathcal{T}]\!]_{\mathfrak{M}}$ is used when one such tree is fixed by context.*

The dominant arithmetic model in practice is **floating-point**: a basic operation $\odot \in \{+, -, \times, \div\}$ is evaluated in real arithmetic and rounded back into FP, yielding a relative error $\delta$ with $\square(a \odot b) = (a \odot b)(1 + \delta)$.

This step breaks associativity. For instance, in IEEE 32-bit floating point, with $a = 1$, $b = 2^{24}$, $c = -2^{24}$, and $\square(1 + 2^{24}) = 2^{24}$:

$$(a + b) + c = 0 \neq 1 = a + (b + c).$$

Similar phenomena arise from overflow and exceptional values (see Appendix G). In particular, once associativity is lost, the arithmetic no longer forms a monoid under $\odot$, meaning that operations that assume this structure (changing parenthesation of evaluations, etc.) are no longer well-defined.

For an overview of arithmetic models used in the expressivity literature, see Appendix G.

### 4.2. Evaluation Order

Non-associativity makes evaluation order semantically relevant: the value of an expression may depend on how it is parenthesized. Any expression built from binary operations induces an evaluation tree specifying a bracketing (and, operationally, an order of intermediate rounding).

In algebraic language theory, letter-by-letter transitions are modeled by function composition, which is associative by definition. To treat an RNN update as a monoid operation when the underlying arithmetic is non-associative, two conditions are required:

1. The evaluation order *within each transition update* must be fixed so that the update map is well-defined on every admissible input.

2. The evaluation of recurrent updates must respect temporal order: the update at time $t$ must be completed before the update at time $t + 1$.

**Definition 4.3** (Recurrence-consistent evaluation). *An evaluation strategy is **recurrence-consistent** if conditions 1 and 2 are met.*

If either condition fails, the model's behavior on sequences is not a well-defined recurrence, and formal statements about languages recognized via recurrent transitions become ill-posed.

Modern hardware and compilers may reorder operations for performance, often relying on assumed associativity, and may even choose evaluation orders that vary across executions. This can violate the conditions above, even when the model is syntactically recurrent.

Discretization and evaluation order are not mere afterthoughts; they fundamentally determine the architecture's expressivity. This can have surprising consequences. For instance, Merrill et al. (2024, Theorem 4.2) implies that RNNs without nonlinearity cannot simulate arbitrary finite machines. However, the theorem assumes a word-level evaluation order that does not satisfy condition 2, meaning it is not recurrence-consistent. It can be shown that, under different implementation semantics with consistent evaluation order, nominally linear recurrences are sufficient to simulate arbitrary finite machines. See Appendix H for details.

## 5. Conclusion

The paper establishes an algebraic characterization of recurrent language models in which the arithmetic semantics is part of the object characterized. For any model with recurrence-consistent evaluation, the global dynamics factor through the wreath product of the layer transition monoids; the syntactic monoid of any recognized language thereby divides this realized wreath product. Expressivity questions reduce to divisibility statements in a family of finite monoids, decidable via Krohn–Rhodes decomposition.

The stratification (cores, wirings, arithmetic) isolates the arithmetic semantics as the single component on which the realized monoids depend. The diagonal-SSM case study makes this concrete: the same idealized recurrence supports no modular counting under nonnegative floating-point arithmetic, yet recognizes parity under unsigned-integer quantization. The result of Sarrof et al. (2024) on the impossibility of parity in nonnegative diagonal SSMs is recovered as an instance of the divisibility chain.

The same algebraic analysis applies, with some adaptation, to attention-based architectures once attention is formalized as a recurrence, a direction we leave for future work.

## Acknowledgements

We would like to thank the anonymous reviewers for their valuable remarks and suggestions.

## Impact Statement

This paper presents theoretical work aimed at advancing the field of machine learning by providing a procedure for understanding the formal capabilities of language models, which may also lead to the discovery of limitations and, in turn, to more capable architectures. Hence, as with any machine learning research, there are many potential societal consequences of our work, none of which we feel must be specifically highlighted here.

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

# Contents

# A. Notation

**Typographic convention.** Upright/italic fonts denote *algebraic objects* (sets, monoids, morphisms, languages: $M, Q, X,$ $\Sigma, \mathbb{W}_\mathscr{R}, f, g$); calligraphic and decorated fonts denote *semantic and machines related objects* ($\mathscr{L}, \mathscr{R}, \mathscr{A}, \mathfrak{c}, \mathfrak{c}_\bullet, \mathscr{T}, \mathfrak{M}$).

**Monoids and transformation monoids.**

| | |
|---|---|
| $M \leq N$ | Submonoid inclusion. |
| $M \prec N$ | Division: $M$ is a quotient of a submonoid of $N$. |
| $\langle S \rangle_{\mathrm{mon}}$ | Submonoid generated by $S$. |
| $Q^Q$ | Full transformation monoid of $Q$. |
| $(M, Q)$ | Transformation monoid: $M \leq Q^Q$. |
| $M(\mathscr{L})$ | Syntactic monoid of $\mathscr{L}$. |
| $\times, \wr$ | Direct product and (left) wreath product. |
| $\mathbb{Z}/n\mathbb{Z}$ | cyclic group of order $n$. |

**Algebraic RNNs.**

| | |
|---|---|
| $\mathfrak{c}$ | Algebraic core $(Q, X, Y, f, g)$. |
| $M_\mathfrak{c}$ | Transition monoid of $\mathfrak{c}$, generated by $\{f_x\}$. |
| $\mathscr{R}$ | Algebraic RNN of depth $N$, with cores $\mathfrak{c}_n$ and wirings $\tau_n$. |
| $Q_\mathscr{R}$ | Global state space $Q_1 \times \cdots \times Q_N$. |
| $\mathscr{R} \rhd_\tau \mathscr{R}'$ | Cascade juxtaposition along $\tau$. |

**Wreath products and realized dynamics.**

| | |
|---|---|
| $\mathbb{W}_\mathscr{R}, W_\mathscr{R}$ | Ambient wreath product and its underlying monoid. |
| $T \subseteq X_1$ | Admissible inputs to the first core. |
| $\varphi_n^T$ | Layer-input dependency map relative to $T$. |
| $M_n^T$ | Realized layer monoid relative to $T$. |
| $\mathbb{W}_\mathscr{R}^T$ | Realized wreath product relative to $T$. |
| $M_\mathscr{R}^T$ | Global transition monoid driven by $T$. |

**Acceptors.**

| | |
|---|---|
| $\mathscr{A}$ | RNN acceptor $(\Sigma, \mathscr{R}^+, q^\circ, e)$. |
| $\mathscr{R}^+$ | Augmented RNN $\mathscr{R} \rhd_{\tau_N} \mathfrak{c}_\bullet$. |
| $\mathfrak{c}_\bullet$ | Accepting core (Moore-style); $F \subseteq Q_\bullet$ the accepting region. |
| $\mathscr{L}(\mathscr{A})$ | Language recognized by $\mathscr{A}$. |

**Arithmetic models.**

| | |
|---|---|
| $\mathfrak{M}$ | Arithmetic model $(\mathscr{D}, \mathscr{O}, \Box)$. |
| $\llbracket f \rrbracket_\mathfrak{M}$ | Evaluation of $f$ under $\mathfrak{M}$. |
| FP | Floating point. |
| $\mathrm{INT}^p$ | Unsigned integers modulo $2^p$. |
| **Ap** | Aperiodic pseudovariety |

## B. A Structural Perspective

The paper's algebraic account sits between two hierarchies the literature has historically kept apart: the *analytic* hierarchy of real-valued recurrent architectures, and the *algebraic* hierarchy of finite-state automata. The bridge between them, discretization, is usually treated as a numerical perturbation absorbed by an $\varepsilon$. This is a category error in the technical sense: it confuses two structurally distinct levels.

A natural reflex is to organize the situation as a functor $\mathscr{F}_{\mathfrak{M}} \colon \mathbb{R}\text{-}\mathbf{RNN} \to \mathbf{Fin}\text{-}\mathbf{RNN}$ from real-valued to finite-state RNNs, parameterized by an arithmetic $\mathfrak{M}$. This picture is wrong structurally: $\mathscr{F}$ does not preserve composition. For state updates $f, g$ over $\mathbb{R}$,

$$\mathscr{F}(g \circ f) \neq \mathscr{F}(g) \circ \mathscr{F}(f),$$

because rounding does not distribute over composition. The local loss of associativity in $\mathfrak{M}$ (Section 4) is the global non-functoriality of $\mathscr{F}$. A theorem in $\mathbb{R}\text{-}\mathbf{RNN}$ that relies on associativity, the chain of compositions underlying Siegelmann–Sontag Turing-completeness (Siegelmann and Sontag, 1992), for instance, does not transfer to $\mathbf{Fin}\text{-}\mathbf{RNN}$ without a new proof carried out in the discrete setting. What survives, once $\mathfrak{M}$ is fixed, is the realized transition monoid $M_{\mathscr{F}(\mathscr{R})}^{e(\Sigma)}$ of Definition 3.10: a finite object carrying its own monoid structure inherited from $\mathfrak{M}$, and the algebraic invariant that any syntactic monoid of a recognized language must divide.

Two consequences follow, each correcting a claim common in the literature. First, real-valued expressivity is not a lower bound on discrete expressivity: the Siegelmann–Sontag construction, discretized underfixed-evaluation floating-point, lands in a strictly subregular pseudovariety. Second, discrete expressivity is not a corruption of real-valued expressivity: wraparound integer arithmetic makes available cyclic group structure that real arithmetic does not naturally realize. Discretization does not project expressivity downward; it maps it onto a different pseudovariety, determined by $\mathfrak{M}$.

Real-valued expressivity is an ideal; discrete expressivity is what computes. The translation between them cannot be analyzed at the level of morphisms in any meaningful category, because the proposed functor is not one. What can be analyzed are the algebraic invariants of $\mathscr{F}(\mathscr{R})$: its realized transition monoid and the pseudovariety it inhabits. These invariants were the object of the present paper, and of the case studies that follow.

## C. Limitations

This paper establishes expressivity rather than learnability: it asks whether there exists a specific parametrization of a given architecture from a family that can recognize a given formal language under a fixed numerical semantics. However, it makes no claims about whether such a parametrization could be found automatically through gradient-based optimization.

The present treatment is restricted to finite-precision semantics. As a result, every induced transition monoid is finite, and every recognized language is at most regular. The construction could be extended to account for settings in which precision or depth grows with input length, thereby yielding infinite monoids and enabling the architecture to capture non-regular languages.

Furthermore, the paper focuses on explicitly recurrent architectures, such as Elman-RNNs and structured state-space models. However, transformers are currently the dominant language model architecture, raising the question of whether the algebraic construction of this paper could be adapted to attention-based architectures as well. To fit the abstraction of transformation monoids established here, transformers would need to be formalized as recurrent models, which is natural for linear attention and certain finite-precision implementations, but requires further work.

Lastly, the paper focuses on establishing a purely algebraic abstraction of recurrent neural models and demonstrates its applicability through a case study of discrete diagonal state-space models. While this rederives and corrects known expressivity results about an important practical architecture, the paper does not attempt to prove fundamentally novel results about the expressivity of the plethora of other sequence-model architectures existing in the literature.

## D. Common Types of RNNs

Recall the definition of an algebraic RNN core.

**Definition 3.1.** *An **algebraic core** is a tuple $\mathfrak{c} = (Q, X, Y, f, g)$ where $Q$, $X$, and $Y$ are sets, and*

$$f \colon Q \times X \to Q, \qquad\qquad g \colon Q \times X \to Y$$

*are total maps, called respectively the **transition** and **readout** functions.*

This definition is broad enough to apply to many RNN architectures.

**Elman RNN.** Let $Q = \mathbb{R}^m$ and $X = Y = \mathbb{R}^d$. With parameters $W_q \in \mathbb{R}^{m \times m}$, $U_q \in \mathbb{R}^{m \times d}$, $W_y \in \mathbb{R}^{d \times m}$, $b_q \in \mathbb{R}^m$, $b_y \in \mathbb{R}^d$ and nonlinearities $\sigma_q$ and $\sigma_y$:

$$f(q_{t-1}, x_t) = \sigma_q(W_q q_{t-1} + U_q x_t + b_q) \tag{D.2}$$
$$g(q_t, x_t) = \sigma_y(W_y q_t + b_y) \tag{D.3}$$

**GRU.** Let $Q = \mathbb{R}^m$ and $X = Y = \mathbb{R}^d$. With sigmoid $\sigma$ and elementwise product $\odot$, parameters $W_{\{z,r,c\}} \in \mathbb{R}^{m \times d}$, $U_{\{z,r,c\}} \in \mathbb{R}^{m \times m}$, $b_{\{z,r,c\}} \in \mathbb{R}^m$, $W_y \in \mathbb{R}^{d \times m}$, and $b_y \in \mathbb{R}^d$:

$$z_t = \sigma(W_z x_t + U_z q_{t-1} + b_z) \tag{D.4}$$
$$r_t = \sigma(W_r x_t + U_r q_{t-1} + b_r) \tag{D.5}$$
$$\tilde{q}_t = \tanh(W_c x_t + U_c(r_t \odot q_{t-1}) + b_c) \tag{D.6}$$
$$\tag{D.7}$$

The recurrence and output are then given by:

$$f(q_{t-1}, x_t) = (1 - z_t) \odot q_{t-1} + z_t \odot \tilde{q}_t \tag{D.8}$$
$$g(q_t, x_t) = \sigma_y(W_y q_t + b_y) \tag{D.9}$$

**LSTM.** Let $Q = \mathbb{R}^m \times \mathbb{R}^m \cong \mathbb{R}^{2m}$ and $X = Y = \mathbb{R}^d$. Write $q_t = (q_t^{(c)}, q_t^{(h)})$ with $q_t^{(c)}, q_t^{(h)} \in \mathbb{R}^m$. With sigmoid $\sigma$, elementwise product $\odot$, parameters $W_{\{i,f,o,g\}} \in \mathbb{R}^{m \times d}$, $U_{\{i,f,o,g\}} \in \mathbb{R}^{m \times m}$, $b_{\{i,f,o,g\}} \in \mathbb{R}^m$, $W_y \in \mathbb{R}^{d \times m}$, and $b_y \in \mathbb{R}^d$:

$$i_t = \sigma(W_i x_t + U_i q_{t-1}^{(h)} + b_i) \tag{D.10}$$
$$f_t = \sigma(W_f x_t + U_f q_{t-1}^{(h)} + b_f) \tag{D.11}$$
$$o_t = \sigma(W_o x_t + U_o q_{t-1}^{(h)} + b_o) \tag{D.12}$$
$$\tilde{q}_t^{(c)} = \tanh(W_g x_t + U_g q_{t-1}^{(h)} + b_g) \tag{D.13}$$

The recurrence and output are then given by:

$$f(q_{t-1}, x_t) = \left( f_t \odot q_{t-1}^{(c)} + i_t \odot \tilde{q}_t^{(c)}, o_t \odot \tanh(f_t \odot q_{t-1}^{(c)} + i_t \odot \tilde{q}_t^{(c)}) \right) \tag{D.14}$$
$$g(q_t, x_t) = \sigma_y(W_y q_t^{(h)} + b_y) \tag{D.15}$$

**SSM.** Let $Q = \mathbb{R}^m$ and $X = Y = \mathbb{R}^d$. With parameters $A \in \mathbb{R}^{m \times m}$, $B \in \mathbb{R}^{m \times d}$, $C \in \mathbb{R}^{d \times m}$, define

$$f(q_{t-1}, x_t) = A q_{t-1} + B x_t \tag{D.16}$$
$$g(q_t, x_t) = C q_t \tag{D.17}$$

In some variants, A, B, and C are themselves functions of the input, meaning

$$f(q_{t-1}, x_t) = A(x_t) q_{t-1} + B(x_t) x_t \tag{D.18}$$
$$g(q_t, x_t) = C(u_t) q_t \tag{D.19}$$

*Remark* D.1. Note that we explicitly add a dependence on the input for both the readout $g$, as well as the wiring function $\tau$. This allows for the optional formalization of residual connections, both for the RNN core alone and for the entire layer.

# E. Proof of the Factorization Lemma

We restate and prove the factorization lemma of Section 3.5. The proof is by induction on the depth $N$ of $\mathcal{R}$ and rest on a single observation: the input $x_{n+1}$ delivered to layer $n + 1$ depends on the states of the lower layers before they update on the current symbol, never after. This is precisely the dependency pattern encoded by the left-wreath-product convention of Section 2, where a wreath element $(m, \phi)$ acts as $(q, q') \mapsto (m \cdot q, \phi(q) \cdot q')$, with $\phi$ evaluated at the old lower state.

**Lemma 3.12** (Factorization). *Let $T \subseteq X_1$. For each $t \in T$, the global state update*

$$(F_{\mathcal{R}})_t \colon Q_{\mathcal{R}} \to Q_{\mathcal{R}}$$

*induced by feeding $t$ into the first core is an element of $W_{\mathcal{R}}^T$. Consequently, the global transition monoid driven by $T$,*

$$M_{\mathcal{R}}^T \overset{\text{def}}{=} \big\langle (F_{\mathcal{R}})_t \mid t \in T \big\rangle_{\text{mon}},$$

*satisfies*

$$M_{\mathcal{R}}^T \leq W_{\mathcal{R}}^T \leq W_{\mathcal{R}}.$$

*Proof.* We induct on $N$.

**Base case, $N = 1$.** An algebraic RNN of depth 1 consists of a single core $\mathfrak{c}_1$ and no wiring maps; its global state space is $Q_{\mathcal{R}} = Q_1$ and its realized wreath product is $\mathbb{W}_{\mathcal{R}}^T = (M_1^T, Q_1)$. For $t \in T$,

$$(F_{\mathcal{R}})_t(q_1) = f_1(q_1, t) = (f_1)_t(q_1).$$

By Definition 3.9, $X_1^T = T$, hence $t \in X_1^T$, hence $(f_1)_t \in M_1^T = W_{\mathcal{R}}^T$ by Definition 3.10.

**Inductive step.** Let $\mathcal{R}$ be of depth $N + 1$ and let $\mathcal{R}|_{\leq N}$ denote its truncation to the first $N$ layers, itself an algebraic RNN, equipped with the same generating set $T \subseteq X_1$. By Definition 3.11,

$$\mathbb{W}_{\mathcal{R}}^T = \mathbb{W}_{\mathcal{R}|_{\leq N}}^T \wr (M_{N+1}^T, Q_{N+1}).$$

Decompose a global state of $\mathcal{R}$ as $(q_{<N+1}, q_{N+1})$ where $q_{<N+1} \overset{\text{def}}{=} (q_1, \ldots, q_N) \in Q_{\mathcal{R}|_{\leq N}}$. Fix $t \in T$. We analyze the two components of $(F_{\mathcal{R}})_t$.

*(a) Lower-layer component.* The wiring of an algebraic RNN is strictly feedforward: by Definition 3.9, the input $x_n$ to layer $n$ depends only on $(q_{<n}, t)$. Consequently the update of the first $N$ layers under symbol $t$ coincides with the global update of $\mathcal{R}|_{\leq N}$:

$$\pi_{<N+1}\big((F_{\mathcal{R}})_t(q_{<N+1}, q_{N+1})\big) = (F_{\mathcal{R}|_{\leq N}})_t(q_{<N+1}),$$

where $\pi_{<N+1}$ is the projection onto the first $N$ coordinates. By the inductive hypothesis applied to $\mathcal{R}|_{\leq N}$ with the same $T$,

$$(F_{\mathcal{R}|_{\leq N}})_t \in W_{\mathcal{R}|_{\leq N}}^T.$$

*(b) Upper-layer component.* The update of layer $N + 1$ is

$$q_{N+1}' = f_{N+1}\big(q_{N+1}, x_{N+1}\big), \quad x_{N+1} = \varphi_{N+1}^T(q_{<N+1}, t).$$

Define

$$\psi_t \colon Q_{\mathcal{R}|_{\leq N}} \longrightarrow M_{N+1}^T, \qquad \psi_t(q_{<N+1}) \overset{\text{def}}{=} (f_{N+1})_{\varphi_{N+1}^T(q_{<N+1}, t)}.$$

*(c)* Combining (a) and (b) shows

$$(F_{\mathcal{R}})_t(q_{<N+1}, q_{N+1}) = \Big((F_{\mathcal{R}|_{\leq N}})_t(q_{<N+1}), \psi_t(q_{<N+1})(q_{N+1})\Big).$$

This is exactly the action on $Q_{\mathcal{R}}$ of the wreath element

$$\big((F_{\mathcal{R}|_{\leq N}})_t, \psi_t\big) \in W_{\mathcal{R}|_{\leq N}}^T \wr M_{N+1}^T = W_{\mathcal{R}}^T,$$

where the evaluation of $\psi_t$ at the *old* lower state $q_{<N+1}$, matches the left-wreath convention precisely because $x_{N+1}$ depends only on the old lower-layer states. Accordingly, $(F_{\mathcal{R}})_t \in W_{\mathcal{R}}^T$, which closes the induction.

**The action of the empty word.** The monoid $M_{\mathcal{R}}^T = \left\langle (F_{\mathcal{R}})_t \mid t \in T \right\rangle_{\mathrm{mon}}$ contains, by definition, the identity $\mathrm{id}_{Q_{\mathcal{R}}}$. Under the morphism $T^* \to M_{\mathcal{R}}^T$, the empty word $\varepsilon \in T^*$ is sent to $\mathrm{id}_{Q_{\mathcal{R}}}$. The same is true of $W_{\mathcal{R}}^T$: its identity is the wreath element $(1_{W_{\mathcal{R}|\leq N}^T}, q \mapsto 1_{M_{N+1}^T})$, which acts as the identity on $Q_{\mathcal{R}}$.

**Conclusion.** We have shown that every generator of $M_{\mathcal{R}}^T$ lies in $W_{\mathcal{R}}^T$, and that the identities of the two monoids agree. Therefore,

$$M_{\mathcal{R}}^T = \left\langle (F_{\mathcal{R}})_t \mid t \in T \right\rangle_{\mathrm{mon}} \leq W_{\mathcal{R}}^T.$$

The remaining inclusion $W_{\mathcal{R}}^T \leq W_{\mathcal{R}}$ is monotonicity of the wreath product: $M_n^T \leq M_n$ for every $n$ (Definition 3.10), and a wreath element $(m, \phi)$ with $m \in M_n^T$ and $\phi$ taking values in $M_{n+1}^T$ is, by Definition 2.8, an element of the ambient wreath product $W_{\mathcal{R}}$. $\square$

*Remark* E.1. The proof above shows the following: if $T \subseteq T'$, then induction on $n$ gives $X_n^T \subseteq X_n^{T'}$ at every layer (the dependency map $\varphi_n^T$ is the restriction of $\varphi_n^{T'}$ to $Q_{<n} \times T$), hence $M_n^T \leq M_n^{T'}$, hence $\mathbb{W}_{\mathcal{R}}^T \leq \mathbb{W}_{\mathcal{R}}^{T'}$, hence $M_{\mathcal{R}}^T \leq M_{\mathcal{R}}^{T'}$.

# F. Proof of the Cascade Juxtaposition as Wreath Extension

**Proposition 3.20.** *Let $\mathcal{R}$ and $\mathcal{R}'$ be algebraic RNNs of depths $N$ and $N'$ respectively, and let $\tau \colon \mathrm{Out}(\mathcal{R}) \to \mathrm{In}(\mathcal{R}')$ be a wiring map between them. For any $T \subseteq \mathrm{In}(\mathcal{R})$,*

$$\mathbb{W}_{\mathcal{R} \triangleright_\tau \mathcal{R}'}^T = \mathbb{W}_{\mathcal{R}}^T \wr \mathbb{W}_{\mathcal{R}'}^T,$$

*where the equality is understood up to canonical isomorphism (Remark 2.10), and $\mathbb{W}_{\mathcal{R}'}^T$ denotes the realized wreath product of $\mathcal{R}'$ relative to the set of inputs reachable through $\tau$ from states of $\mathcal{R}$ activated by $T$. The realized transition monoid of the juxtaposition satisfies the chain*

$$M_{\mathcal{R}}^T \prec M_{\mathcal{R} \triangleright_\tau \mathcal{R}'}^T \leq W_{\mathcal{R}}^T \wr W_{\mathcal{R}'}^T,$$

*the first division being given by the canonical projection $\pi_{\leq N} \colon Q_{\mathcal{R} \triangleright_\tau \mathcal{R}'} \to Q_{\mathcal{R}}$.*

*Proof.* Throughout the proof, we write $\mathcal{R}'' \overset{\mathrm{def}}{=} \mathcal{R} \triangleright_\tau \mathcal{R}'$ for brevity, and abbreviate $\cdot$ for $\triangleright_\tau$, the wiring being fixed by hypothesis.

**The wreath identity.** By Definition 3.5, $\mathcal{R}''$ is the algebraic RNN of depth $N + N'$ whose sequence of pairs is the concatenation of those of $\mathcal{R}$ and $\mathcal{R}'$, with $\tau$ inserted as the wiring of the first core of $\mathcal{R}'$. The realized wreath product $W_{\mathcal{R}''}^T$ is then, by Definition 3.11, the iterated wreath product

$$\mathbb{W}_{\mathcal{R}''}^T = (M_1^T, Q_1) \wr (M_2^T, Q_2) \wr \cdots \wr (M_{N+N'}^T, Q_{N+N'}).$$

By Remark 2.10, the wreath product of transformation monoids is associative up to canonical isomorphism. Grouping the first $N$ factors and the last $N'$ factors yields

$$\mathbb{W}_{\mathcal{R}''}^T \cong \left( (M_1^T, Q_1) \wr \cdots \wr (M_N^T, Q_N) \right) \wr \left( (M_{N+1}^T, Q_{N+1}) \wr \cdots \wr (M_{N+N'}^T, Q_{N+N'}) \right).$$

The left bracket is $W_{\mathcal{R}}^T$ by Definition 3.11 applied to $\mathcal{R}$ and the set $T$. The right bracket is the realized wreath product of $\mathcal{R}'$ relative to the inputs delivered by $\tau$ through the cascade, that is, $W_{\mathcal{R}'}^T$ in the sense of the proposition's statement. Hence the asserted identity, modulo the canonical isomorphism.

**The inclusion in the wreath product.** The inclusion $M_{\mathcal{R}''}^T \leq W_{\mathcal{R}''}^T = W_{\mathcal{R}}^T \wr W_{\mathcal{R}'}^T$ is Lemma 3.12 applied to $\mathcal{R}''$ with generating set $T$.

**The division.** For the division $M_{\mathcal{R}}^T \prec M_{\mathcal{R}''}^T$, consider the projection $\pi_{\leq N} \colon Q_{\mathcal{R}''} \to Q_{\mathcal{R}}$ onto the first $N$ coordinates of the global state. The strictly feedforward wiring of the cascade (Definition 3.9) ensures that the dynamics on the first $N$ coordinates are independent of the remaining $N'$ coordinates: for every $t \in T$ and every $(q_{\leq N}, q_{>N}) \in Q_{\mathcal{R}''}$,

$$\pi_{\leq N}\big( (F_{\mathcal{R}''})_t(q_{\leq N}, q_{>N}) \big) = (F_{\mathcal{R}})_t(q_{\leq N}).$$

The projection therefore induces a surjective monoid morphism $\pi_{\leq N}^* \colon M_{\mathcal{R}''}^T \twoheadrightarrow M_{\mathcal{R}}^T$, exhibiting $M_{\mathcal{R}}^T$ as a quotient of $M_{\mathcal{R}''}^T$. This is the asserted division. $\square$

*Table 1.* Properties of main arithmetic models. Memory scaling is given as a function of the position $t$ in the input sequence.

| Model $\mathfrak{M}$ | Domain $\mathcal{D}$ | Operations $\mathcal{O}$ | Overflow | Memory Scaling | Associative |
|---|---|---|---|---|---|
| Reals | $\mathbb{R}$ | $\{+, -, \times, \div\}$ | N.A. | $\infty$ | yes |
| Rationals | $\mathbb{Q}$ | $\{+, -, \times, \div\}$ | N.A. | $\mathcal{O}(t)$ | yes |
| Integers | $\mathbb{Z}$ | $\{+, -, \times\}$ | N.A. | $\mathcal{O}(t)$ | yes |
| Float | $\mathrm{FP} \cup \{\pm\infty, \mathrm{NaN}\}$ | rd. $\{+, -, \times, \div\}$ | $\pm\infty$ | $\mathcal{O}(1)$ | no |
| Int | finite interval of $\mathbb{Z}$ | $\{+, -, \times\}$ | clamp | $\mathcal{O}(1)$ | no |
| UInt | $\mathbb{Z}/2^p\mathbb{Z}$ | $\{+, -, \times\}$ | wrap | $\mathcal{O}(1)$ | yes |

# G. Models of Arithmetic

In the following, we outline the main arithmetic models used to analyze the expressivity of neural architectures. Critically, we point out the tradeoffs between space requirements and the cost of losing algebraic properties (most importantly, associativity).

## G.1. Reals and Rationals

One common choice for expressivity analysis is to assume that all values lie in a field such as the real numbers $\mathbb{R}$ or the rationals $\mathbb{Q}$. This may not be a realistic model for actual neural network implementations, because a given real number may not be representable in finite memory, and rationals may also have non-terminating binary expansions. Note that any specific rational number can be represented with finite space via an integer enumerator and denominator.

## G.2. Floating-Point Arithmetic

By far the most common arithmetic model used to approximate real numbers by computer hardware is floating-point arithmetic. In this model, each of a finite subset of the real numbers is represented as the product of a sign $\pm$, a significand $m$, and an integer power of a base $b$ (usually 2).[2] The arithmetic model is specified by the precision $p$ (the number of digits in the significand) and an integer exponent range $[e_{\min}, e_{\max}]$.

The significand $m$ is a fractional value composed of $p$ digits $(d_0, d_1, \ldots, d_{p-1})$ in base $b$. Ignoring special values for the moment, floating-point numbers have the form:

$$\pm \underbrace{d_0.d_1 d_2 \ldots d_{p-1}}_{\text{significand}} \times \underbrace{b}_{\text{base}}{}^{\overbrace{e}^{\text{exponent}}}$$

where $e_{\min} \leq e \leq e_{\max}$ and $0 \leq d_i < b$ for all $i < p$. A floating-point number is called **normal** if $d_0 \neq 0$. In binary ($b = 2$), this implies $d_0 = 1$. Otherwise, it is called a **subnormal number**, if $d_0 = 0$ and the exponent is fixed at $e = e_{\min}$. Subnormals extend the representable set of numbers toward 0 by trading significant bits of the mantissa for more space to represent smaller exponents.[3]

**Overflow and error behavior.** With any numerical system based on a finite set, an important question is what happens when operations result in values outside that range. In floating-point, numbers whose magnitude is larger than the maximum/minimum representable value result in overflow of dedicated **infinity** values ($+\infty$ and $-\infty$). Adding or multiplying any normal or subnormal floating-point number with $\pm\infty$ yields $\pm\infty$.

For results of invalid computations such as $0/0$, $\infty - \infty$, etc. floating-point has **NaNs** (Not-a-Number). Note that this can also result from the denominator becoming 0 due to underflow. NaNs propagate, meaning that if at any point in a computation a subresult is NaN, the result of the whole computation will be NaN.

---

[2] Note that only numbers whose base-$b$ expansion terminates are exactly representable in that base with finitely many digits. For example, the fraction $1/3$ cannot be exactly represented in base 2.

[3] Note that in floating-point, 0 is not unique, since both $+0$ and $-0$ are valid distinct floating-point numbers.

**Rounding.** For any real number $x$, let $\Box(x)$ denote its floating-point rounding to the nearest representable value.[4] Rounding introduces a small relative error with respect to the ambient system ($\mathbb{R}$) in which floating-point numbers are embedded when the result falls within the normal range. A standard abstraction for $x \in \mathbb{R}$ is:

$$\Box(x) = x(1 + \delta), \qquad |\delta| \leq u,$$

where $u$ is the *unit roundoff* ($u = \frac{1}{2}b^{1-p}$ for round-to-nearest). The spacing between adjacent representable numbers grows with $|x|$, so floating point has roughly constant *relative* precision across magnitudes.

Floating-point arithmetic implements the basic operations $\{+, -, \times, \div\}$ as if computing the real result and then rounding it back into the format with some rounding error $\delta$:

$$\Box(a \odot b) = (a \odot b)(1 + \delta), \quad \odot \in \{+, -, \times, \div\}$$

In typical implementations, rounding occurs after each primitive operation; long multi-term computations therefore accumulate rounding error over many steps.

**Rounding and overflow break associativity.** Rounding is fundamentally at odds with associativity, because the results of computations can differ depending on their order. This happens, for example, when a subterm adds two numbers, one of which is much smaller than the other, so the smaller one disappears. This is called **absorption**.

**Example G.1** (Absorption). *Consider $x = 2^{24} - 2^{24} + 1 = 1$. In 32-bit floating-point, depending on ordering, this yields:*

*(i)* $\Box(\Box(2^{24} - 2^{24}) + 1) = \Box(0 + 1) = 1$

*(ii)* $\Box(2^{24} - \Box(2^{24} + 1)) = \Box(2^{24} - 2^{24}) = 0$

Another reason for associativity to fail in floating-point is overflow and error behavior.

**Example G.2** (Overflow). *Let $f_{\max}$ be the largest representable number smaller than $\infty$ in a given floating-point arithmetic. Then $x = f_{\max} + 1 - 1$ evaluates to:*

*(i)* $\Box(\Box(f_{\max} + 1) - 1) = \Box(\infty + 1) = \infty$

*(ii)* $\Box(f_{\max} + \Box(1 - 1)) = \Box(f_{\max} + 0) = f_{\max}$

For similar reasons, rounding also breaks distributivity, meaning that in general:

$$\Box((x \cdot \Box(y + z)) \neq \Box(\Box(x \cdot y) + \Box(x \cdot z))$$

**Definition G.3** (Magma). *A **magma** is a set $S$ closed under a binary operation $\cdot$, **not** necessarily associative.*

The algebraic structure formed by fixed-precision floating-point arithmetic is an ordered magma, an algebraic object with almost no structure.[5]

**Definition G.4** (Ordered magma). *An **ordered magma** is a magma $(S, \cdot)$ together with a partial order $\leq$ on $S$ such that the binary operation is monotone in both arguments: for all $a, b, c \in S$,*

$$a \leq b \implies a \cdot c \leq b \cdot c \quad and \quad a \leq b \implies c \cdot a \leq c \cdot b.$$

*Equivalently, $\cdot : S \times S \to S$ is order-preserving with respect to the product order on $S \times S$.*

In fact, in order for composite computations in a magma to be well-defined at all, we need to fix an **execution order**, such as always computing terms of the same precedence from left to right.

---

[4]Other rounding modes exist (toward 0, toward $\pm\infty$, stochastic rounding, etc.). The dominant default in scientific computing is round-to-nearest, with ties broken by the ties-to-even rule.

[5]More specifically, it is an ordered commutative unital magma for $+$ and $\times$ and an ordered right-unital magma for $-$ and $\div$ when disregarding NaNs. Note that ordering is not strict due to absorption.

*Remark* G.5. Most expressivity analyses in the literature treat fixed-precision arithmetic, such as floating-point arithmetic, as essentially "Real arithmetic plus $\epsilon$", where all analyses are done over $\mathbb{R}$ and a small error is accounted for in the end. This *does not work* for formal language theory results, because rounding errors cannot be kept arbitrarily small in the general case (e.g., catastrophic cancellation, Goldberg (1991)). Furthermore, formal language theory handles arbitrarily long sequences, which means error accumulation can grow arbitrarily large for any non-zero marginal error.

Note also that the failure of distributivity makes matrix multiplication non-associative in floating-point arithmetic, meaning that regroupings of matrix- or tensor products for computational efficiency will generally not be arithmetically equivalent.

### G.3. Integers

Another common model is integer arithmetic (quantization), which has been used successfully for efficient RNN inference (Jacob et al., 2018; Kim et al., 2021; Sari et al., 2021) and training (Wang et al., 2022; Nia et al., 2023). In contrast to floating-point arithmetic, integer-only arithmetic represents numbers as fixed-width bit words that map directly to the set of integers $\mathbb{Z}$. In this model, a value $x \in \mathbb{Z}$ is represented by a bit word of length $p$ (the precision). Most modern hardware supports $p \in \{8, 16, 32, 64\}$.

**Overflow behavior.** For fixed-size integers, let $\bigcirc$ denote the map that sends any integer to the bounded interval of representable values. For a fixed precision, integer arithmetic on standard hardware typically follows the rules of modular arithmetic, i.e., overflow is handled by **wrap-around**. In this case, for a precision $p$, the result of an operation $\odot \in \{+, -, \times\}$ is:

$$\bigcirc(x \odot y) = x \odot y \pmod{2^p}$$

A common alternative to handle overflow in integer arithmetic, especially in machine learning, is to **clamp** to the lowest value $i_{\min}$ or the highest value $i_{\max}$ at the boundaries of the interval.

$$\bigcirc(x \odot y) = \min(\max(x \odot y, i_{\min}), i_{\max})$$

**Associativity.** The algebraic consequences of overflow handling in integer arithmetic depend sharply on whether wrap-around or clamping is used, and on whether the integers are signed or unsigned.

Wrap-around induces exact modular arithmetic over $\mathbb{Z}/2^p\mathbb{Z}$, where $p$ is the precision. In this case, addition and multiplication are associative and distributive, yielding a **commutative ring** (for addition and multiplication) and hence a well-structured algebra.

Clamped arithmetic behaves fundamentally differently. For signed integers, clamping breaks associativity:

$$\bigcirc(\bigcirc(x \odot y) \odot z) \neq \bigcirc(x \odot \bigcirc(y \odot z))$$

in general, because intermediate results may saturate at $i_{\min}$ or $i_{\max}$ depending on evaluation order. Distributivity over multiplication also fails for the same reason. The resulting structure is only a **commutative ordered unital bimagma**: closure and commutativity hold, but associativity and distributivity do not.

For unsigned integers, clamping preserves associativity of addition and multiplication, since saturation occurs only at a single boundary and cannot be crossed in the opposite direction. In this case, clamped arithmetic forms a **commutative ordered bimonoid**: associativity and units are preserved, but inverses are absent. In this case, the resulting algebra is a strictly ordered commutative magma.

Crucially, even when scalar associativity is preserved (as in unsigned clamped arithmetic), matrix associativity fails in all clamped settings. Matrix multiplication relies on distributivity to rearrange sums of products; clamping inside the scalar operations breaks this property, so in general

$$(AB)C \neq A(BC)$$

As a result, clamped integer arithmetic does not admit a well-defined matrix monoid without fixing an execution order, mirroring the situation for floating-point arithmetic.

**Unbounded Integers** Modern programming languages like Python include integer datatypes that can grow arbitrarily large. This means there is no possibility for overflow and hence no need to define wrap-around or saturation behavior. The resulting algebraic structure is simply the ring of integers $\mathbb{Z}$. Note that unbounded integer representations of recurrent hidden states can grow linearly in the sequence length.

## G.4. Fixed-Point Precision

Another common model is **fixed-point** arithmetic. We will not focus on it here, as it does not introduce unique algebraic structures and is effectively an intermediate between floating-point and integer arithmetic. In this model, each number is represented as a scaled integer, with the radix point (decimal or binary) fixed in a consistent position across all values and operations. Results are rounded or truncated to ensure they remain within the representable set. Consequently, fixed-point inherits the rounding complexities and non-associativity of floating-point arithmetic; however, these issues are often more pronounced because the rounding error is absolute (constant) rather than relative to the magnitude of the value. Ultimately, it can be viewed as an extension of integer arithmetic, in which the unit of least precision is shifted by a fixed scaling factor.

# H. Example: Clamped Linear Recurrence

The written recurrence of an RNN layer alone does not determine the implemented transition map. In particular, a source-level affine update may become nonlinear once finite-precision semantics are fixed. This is not merely a pathological possibility: in standard fixed-point and integer-only implementations, matrix products are typically accumulated in a wider register and then requantized or saturated when cast back to the target format, so clamping already appears as part of realistic execution semantics (Jacob et al., 2018; Kim et al., 2021). The crucial issue is therefore not whether clamping occurs, but where the clamp barriers occur in the staged implementation.

To make this explicit, compare the usual Elman recurrence (Eq. (D.2)), where we choose ReLU for $\sigma_q$

$$f(q_{t-1}, x_t) = \text{ReLU}(W_q q_{t-1} + U_q x_t + b_q), \tag{H.20}$$

with the source-level affine update

$$f(q_{t-1}, x_t) = A q_{t-1} + B x_t + b_1 + b_2 + b_3, \tag{H.21}$$

where $q_{t-1}, q_t, b_i \in \mathbb{R}^d$ for $0 \le i \le 3, d \in \mathbb{R}^+, x_t \in \mathbb{R}^n, W_q, A \in \mathbb{R}^{d \times d}, U_q, B \in \mathbb{Z}^{d \times n}$, and $t, n, d, q \in \mathbb{Z}^+$.

Syntactically, the latter update contains no nonlinearity. Now fix a consistent evaluation order and a staged clamped signed-integer semantics with coordinatewise saturation

$$\bigcirc(z) = \max(\min(z, c), -c), \tag{H.22}$$

for some constant $c \in \mathbb{Z}^+$. So, all values are integers and, at chosen points of evaluation, are clamped back to the range of allowed values $[-c, c] \subseteq \mathbb{Z}$ using Eq. (H.22).

For the affine update of Eq. (H.21), we fix the following evaluation semantics: Assume that the affine term $A q_{t-1} + B x_t + b_1$ is first materialized in the target fixed integer format, and that the subsequent bias additions are then each performed successively in that same clamped format. The full implemented update is therefore:

$$q_t = \bigcirc\Big(\bigcirc\big(\bigcirc(A q_{t-1} + B x_t + b_1) + b_2\big) + b_3\Big).$$

The intermediate clamp barriers are part of the semantics. The Elman update of Eq. (H.20), on the other hand, is computed with a single clamping step as

$$q_t = \bigcirc\big(\text{ReLU}(W_q q_{t-1} + U_q x_t + b_q)\big),$$

**Proposition H.1.** *Let the recurrent update be as in Eq. (H.20). Choose $A = W_q, B = U_q, b_1 = b_q, b_2 = -c\mathbf{1}_d$, and $b_3 = c\mathbf{1}_d$, where $\mathbf{1}_d$ is a d-dimensional vector of 1s, and c is the constant from Eq. (H.22).*

*Let $v = W_q q_{t-1} + U_q x_t + b_q$. If $v \in [-c, c]^d$, then the staged clamped affine recurrence computes $q_t = \bigcirc\big(ReLU(v)\big)$ exactly, coordinatewise.*

*Proof.* Fix a coordinate $1 \le i \le d$. Since $v_i \in [-c, c]$, the first clamp is inactive, so $\bigcirc(v_i) = v_i$. At the second stage,

$$\bigcirc(v_i - c) = \begin{cases} -c & \text{if } v_i \le 0, \\ v_i - c & \text{if } v_i > 0, \end{cases}$$

because $v_i - c \leq -c$ in the first case and $v_i - c \in (-c, 0]$ in the second. After adding $c$ and clamping once more, we obtain

$$\bigcirc(\bigcirc(v_i - c) + c) = \begin{cases} 0 & \text{if } v_i \leq 0, \\ v_i & \text{if } v_i > 0. \end{cases}$$

Hence the output is $\max(v_i, 0) = \bigcirc(\text{ReLU}(v_i))$. $\square$

Intuitively, the gadget "subtract $c$, clamp, add $c$" turns the lower saturation boundary into a zeroing threshold: negative values are pushed to $-c$ and return as $0$, while positive values remain in the unsaturated regime and survive unchanged.

**Corollary H.2.** *Any DFA simulation theorem for ReLU Elman recurrences using bounded-range integers transfers to this staged clamped affine model.*

*Proof.* Classical threshold-network constructions, dating back to Minsky, show how finite dynamics can be embedded in recurrent networks via thresholding. For the bounded finite-precision setting relevant here, Korsky and Berwick (2019) show constructively that finite-precision single-layer ReLU recurrences are exactly as computationally powerful as deterministic finite automata, using only integers in a bounded range. By Proposition H.1, each such ReLU update can be replaced, within the same bounded working region, by the staged clamped affine update above. Therefore, the same hidden-state transition system, and hence the same DFA simulation, is realized. $\square$

Note that this only apparently contradicts prior limitations for linear recurrences (e.g., Merrill et al., 2024, Theorem 4.2). Those results assume computational semantics under which the recurrence remains genuinely linear and also follow a recurrence-consistent evaluation strategy. By contrast, we show that whether a recurrence is genuinely linear cannot be read off from the written formula alone. Once finite-precision clamping is part of the operational semantics, a nominally affine recurrence may already contain the thresholding nonlinearity on which classical automata simulations rely.

## I. Krohn–Rhodes Theorem

Here, we briefly state a transformation-monoid version of the Krohn–Rhodes theorem (Krohn and Rhodes, 1965).

We first recall the two kinds of prime transformation monoids that occur in the decomposition.

**Definition I.1.** *The **flip-flop** is the transformation monoid*

$$\text{FF} \stackrel{\text{def}}{=} (U_3, \{q_a, q_b\}),$$

*where $U_3 \stackrel{\text{def}}{=} \{\text{Id}, a, b\}$ consists of the identity transformation together with the two constant transformations*

$$a(q) = q_a, \qquad b(q) = q_b, \qquad \forall q \in \{q_a, q_b\}.$$

*Its transition monoid has the following multiplication table:*

| $U_3$ | Id | $a$ | $b$ |
|-------|-----|-----|-----|
| Id | Id | $a$ | $b$ |
| $a$ | $a$ | $a$ | $b$ |
| $b$ | $b$ | $a$ | $b$ |

*where composition is read in the convention used throughout this paper.*

**Definition I.2.** *A **prime transformation group** is a transformation monoid $(G, G)$ where $G$ is a nontrivial finite simple group acting faithfully on itself by right multiplication.*

Thus, the prime transformation groups are exactly

$$\{(G, G) \mid G \text{ is a nontrivial finite simple group}\}.$$

Equivalently, the possible groups $G$ are the non-abelian finite simple groups together with the cyclic groups $\mathbb{Z}/p\mathbb{Z}$ of prime order $p$.

**Theorem I.3** (Krohn–Rhodes). *Let $(M, Q)$ be a finite transformation monoid. Then there exist transformation monoids*

$$(M_1, Q_1), \ldots, (M_k, Q_k)$$

*such that each $(M_i, Q_i)$ is either the flip-flop* FF *or a prime transformation group $(G_i, G_i)$, and*

$$(M, X) \prec (M_1, Q_1) \wr (M_2, Q_2) \wr \cdots \wr (M_k, Q_k).$$

*That is, every finite transformation monoid divides an iterated wreath product of flip-flops and prime transformation groups.*

## J. Case Study: Diagonal State Space Models

This section instantiates the algebraic theory of Section 3 on diagonal SSMs with input-dependent transitions, in the spirit of Sarrof et al. (2024). The same idealized recurrence is analyzed under several arithmetic models, and the resulting expressivity differs from one model to the next.

### J.1. Architecture specification

The diagonal SSM is an algebraic RNN in the sense of Definition 3.4. To make the translation between the two transparent, the symbols of the algebraic core (Definition 3.1) are reused verbatim: the state, input, and readout sets are $Q_n, X_n, Y_n$, the transition and readout maps are $f_n, g_n$, and the wiring is $\tau_n$. Every algebraic statement of Section 3 thus translates to the present setting by direct substitution.

Fix integers $N, m, d \in \mathbb{Z}^+$, where $N$ is the depth, $m$ the hidden dimension, and $d$ the model dimension. For each layer $n \in \{1, \ldots, N\}$, set $Q_n = \mathbb{R}^m$ and $X_n = Y_n = \mathbb{R}^d$. Write $\mathrm{Diag}_m \subseteq \mathbb{R}^{m \times m}$ for the ring of diagonal $m \times m$ real matrices, and identify a diagonal matrix with its diagonal vector, $\mathrm{Diag}_m \cong \mathbb{R}^m$. Write $\mathrm{Diag}_m^+ \cong \mathrm{Diag}_m$ for the sub-semiring of diagonals with nonnegative values.

The layer is parameterized by three learned maps, not assumed linear,

$$A_n \colon \mathbb{R}^d \to \mathrm{Diag}_m, \qquad B_n \colon \mathbb{R}^d \to \mathbb{R}^{m \times d}, \qquad C_n \colon \mathbb{R}^d \to \mathbb{R}^{d \times m},$$

assigning to each input the diagonal transition matrix, the input matrix, and the readout matrix. The recurrence and readout maps $f_n \colon Q_n \times X_n \to Q_n$ and $g_n \colon Q_n \times X_n \to Y_n$ are, for every $t \geq 1$, every $q_{t-1} \in \mathbb{R}^m$, and every $x_t \in \mathbb{R}^d$,

$$f_n(q_{t-1}, x_t) = A_n(x_t)\, q_{t-1} + B_n(x_t)\, x_t, \tag{J.23}$$
$$g_n(q_t, x_t) = C_n(x_t)\, q_t. \tag{J.24}$$

The wiring map $\tau_n \colon X_n \times Y_n \to X_{n+1}$ is, for $x \in \mathbb{R}^d$ and $y \in \mathbb{R}^d$,

$$\tau_n(x, y) = \mathrm{norm}\big(\mathrm{ReLU}(x\, M)\, y\big) + x,$$

where $M \in \mathbb{R}^{d \times d}$ is a parameter, $\epsilon \in \mathbb{R}_{>0}$ is a fixed stabilization constant, and, for $1 \leq i \leq d$,

$$\mathrm{ReLU}(x)_i = \max(0, x_i), \qquad \mathrm{norm}(x)_i = \frac{x_i}{\sqrt{\frac{1}{d} \sum_{j=1}^d x_j^2 + \epsilon}}.$$

The encoder and decoder are set maps $e \colon \Sigma \to \mathbb{R}^d$ and $d \colon \mathbb{R}^d \to \{0, 1\}$, where $\Sigma$ is the finite input alphabet. The encoder factors through the one-hot embedding $\Sigma \hookrightarrow \mathbb{R}^{|\Sigma|}$ followed by a linear map $\mathbb{R}^{|\Sigma|} \to \mathbb{R}^d$; this is a factorization, not a linearity assumption on $e$, since $\Sigma$ carries no vector-space structure.

**Definition J.1** (Nonnegative diagonal SSM). *A diagonal SSM is **nonnegative** if every transition matrix it produces has nonnegative entries, that is, $A_n \colon \mathbb{R}^d \to \mathrm{Diag}_m^+$, for every $n$.*[6]

---

[6]This is close to Mamba, with ReLU in place of SiLU and no convolution across time.

## J.2. Discretization

The expressivity question is well-posed only after the numerical semantics that turns each symbolic recurrence into a total function has been fixed. Fix an arithmetic model $\mathfrak{M}$ (Definition 4.2) with finite representable set $Q_{\mathfrak{M}} \subseteq \mathbb{R}$, so that the layer state set is $Q_n^{\mathfrak{M}} = Q_{\mathfrak{M}}^m$ and the encoder takes values in $Q_{\mathfrak{M}}^d$. For diagonal SSM updates, the evaluation tree of Eq. (J.23) is fixed coordinatewise (Definition 4.3): at coordinate $i$, the multiplication $A_n(x)_{i,i} q_i$ is evaluated before the addition with the precomputed constant $b_i^{\mathfrak{M}}(x) \stackrel{\text{def}}{=} [\![B_n(x)\,x]\!]_{\mathfrak{M}}\big|_i \in Q_{\mathfrak{M}}$, and Definition 4.2 prescribes a rounding at each of these two nodes:

$$[\![f_n(q,x)]\!]_{\mathfrak{M}}\big|_i = \square\big(\square(A_n(x)_{i,i}\,q_i) + b_i^{\mathfrak{M}}(x)\big).$$

The induced map

$$(F_n)_x^{\mathfrak{M}}: Q_n^{\mathfrak{M}} \to Q_n^{\mathfrak{M}}, \qquad (F_n)_x^{\mathfrak{M}}(q) \stackrel{\text{def}}{=} [\![f_n(q,x)]\!]_{\mathfrak{M}},$$

is total for every admissible input $x \in X_n^{\mathfrak{M}} \stackrel{\text{def}}{=} e(\Sigma) \subseteq Q_{\mathfrak{M}}^d$, on the standing assumption that every realizable parameter choice yields a self-map of $Q_{\mathfrak{M}}$, producing in particular no NaN. The core transition monoid is

$$M_n^{\mathfrak{M}} \stackrel{\text{def}}{=} \big\langle (F_n)_x^{\mathfrak{M}} : x \in X_n^{\mathfrak{M}} \big\rangle_{\text{mon}} \leq (Q_n^{\mathfrak{M}})^{Q_n^{\mathfrak{M}}},$$

and the composition of two steps with inputs $x_1$, $x_2$ coincides with the function composition $(F_n)_{x_2}^{\mathfrak{M}} \circ (F_n)_{x_1}^{\mathfrak{M}}$.

**Arithmetic models.** Two models are used below. FP denotes a fixed finite floating-point format of IEEE type, with its induced finite set of representable values. $\text{INT}^p$ denotes unsigned integer arithmetic with wraparound at $p$ bits, that is, the ring $\mathbb{Z}/2^p\mathbb{Z}$.

**Reduction to one coordinate.** Since $A_n(x)$ is diagonal, the update acts coordinatewise: for every $x \in X_n^{\mathfrak{M}}$ and $1 \leq i \leq m$, the $i$-th coordinate evolves under the one-dimensional affine self-map

$$F_{a,b}^{\mathfrak{M}}: Q_{\mathfrak{M}} \to Q_{\mathfrak{M}}, \qquad q \mapsto \square\big(\square(aq) + b\big),$$

with $a \stackrel{\text{def}}{=} A_n(x)_{i,i} \in Q_{\mathfrak{M}}$ and $b \stackrel{\text{def}}{=} b_i^{\mathfrak{M}}(x) \in Q_{\mathfrak{M}}$. Let $\Pi_i \stackrel{\text{def}}{=} \{(A_n(x)_{i,i}, b_i^{\mathfrak{M}}(x)) : x \in X_n^{\mathfrak{M}}\} \subseteq Q_{\mathfrak{M}} \times Q_{\mathfrak{M}}$ collect the parameter pairs realizable at coordinate $i$, and let $M_i \stackrel{\text{def}}{=} \big\langle F_{a,b}^{\mathfrak{M}} : (a,b) \in \Pi_i \big\rangle_{\text{mon}} \leq Q_{\mathfrak{M}}^{Q_{\mathfrak{M}}}$ be the monoid they generate. The coordinatewise action embeds the core monoid in the product, $M_n^{\mathfrak{M}} \leq M_1 \times \cdots \times M_m$.

The one-dimensional analysis serves two distinct purposes. To exhibit a group inside a core, it suffices to display a single pair $(a,b) \in \Pi_i$, frozen on the other coordinates; this is an existence statement and uses a single generator (Lemmas J.5 and J.9). To bound the group content of a core, one must control the whole monoid $M_i$, hence all words in its generators; the bound holds word by word because the governing invariant, monotonicity or the parity of the number of negative multipliers, is stable under composition (Lemmas J.3 and J.6).

### J.2.1. FLOATING POINT

**Definition J.2.** *A finite monoid $M$ is **aperiodic** if there exists $N \in \mathbb{N}$ with $m^N = m^{N+1}$ for all $m \in M$. The class of finite aperiodic monoids is the pseudovariety $\mathbf{Ap}$.*[7]

**Lemma J.3** (Nonnegative floating-point updates are aperiodic). *Fix $\mathfrak{M} = \text{FP}$ with NaNs excluded and a recurrence-consistent evaluation order. If every realizable one-dimensional update at a coordinate has a nonnegative multiplier, then the monoid $M_i$ it generates is aperiodic.*

*Proof.* Let $(Q_{\text{FP}}, \leq)$ be the finite set of IEEE 754 values, including $\pm\infty$ and excluding NaNs, with its usual total order. Each generator has the form $F_{a,b}^{\text{FP}}(q) = \square(\square(aq) + b)$ with $a \geq 0$. IEEE 754 rounding preserves monotonicity for addition and multiplication under the fixed semantics (Mikaitis, 2024, Thms. II.1, II.3), so multiplication by $a \geq 0$, addition of $b$, and both roundings are order-preserving, whence each generator is an order-preserving self-map of the finite chain. A composition of order-preserving maps is order-preserving, so every $m \in M_i$ is order-preserving. An order-preserving self-map of a finite chain has no nontrivial cycle: for any $q$, the sequence $(m^k \cdot q)_{k \geq 1}$ is monotone in a finite set, hence stabilizes after at most $N = |Q_{\text{FP}}|$ steps, giving $m^N = m^{N+1}$. Therefore $M_i \in \mathbf{Ap}$. $\square$

---

[7] For a definition of pseudovarieties, see Eilenberg and Schützenberger (1976).

**Corollary J.4** (Nonnegative floating-point cores are aperiodic). *Every nonnegative diagonal SSM core over* FP*, with NaNs excluded and fixed evaluation semantics, has an aperiodic transition monoid.*

*Proof.* By Lemma J.3, each coordinate monoid $M_i$ is aperiodic, since the realizable multipliers $A_n(x)_{i,i}$ are nonnegative. The core monoid embeds in $M_1 \times \cdots \times M_m$ by the coordinatewise action. Finite products of aperiodic monoids are aperiodic, and submonoids inherit aperiodicity. $\square$

**Lemma J.5** (Signed floating-point cores contain $\mathbb{Z}/2\mathbb{Z}$). *A diagonal SSM core over* FP *whose realizable multipliers include* $-1$ *contains a transition group isomorphic to* $\mathbb{Z}/2\mathbb{Z}$.

*Proof.* The set $Q_{\mathrm{FP}}$ contains the negation-invariant subset $\{-1, 0, 1\}$, on which negation is exact (treating $+0$ and $-0$ as equal under $\leq$). The one-dimensional update with parameters $a = -1$, $b = 0$ then satisfies $F^{\mathrm{FP}}_{-1,0} \circ F^{\mathrm{FP}}_{-1,0} = \mathrm{Id}$ on $\{-1, 0, 1\}$, so $F^{\mathrm{FP}}_{-1,0}$ has order 2 there, while $F^{\mathrm{FP}}_{1,0} = \mathrm{Id}$. Hence $\langle F^{\mathrm{FP}}_{-1,0}, F^{\mathrm{FP}}_{1,0} \rangle_{\mathrm{mon}} \cong \mathbb{Z}/2\mathbb{Z}$, embedded in the core monoid by freezing the remaining coordinates. $\square$

**Lemma J.6** (Group content of floating-point cores). *Let $M_i$ be the monoid generated by one-dimensional affine floating-point updates at a coordinate, with NaNs excluded. Every subgroup of $M_i$ is either trivial or isomorphic to* $\mathbb{Z}/2\mathbb{Z}$.

*Proof.* The governing invariant is order. For $a > 0$ the map $F^{\mathrm{FP}}_{a,b}(q) = \square(\square(aq) + b)$ is order-preserving, and for $a < 0$ it is order-reversing, under the fixed semantics. Hence, a word in the generators is order-preserving or order-reversing according to the parity of the number of negative multipliers it contains, and every element of $M_i$ is of one of these two kinds. Let $G \leq M_i$ be a subgroup with identity idempotent $e$; then $G$ acts faithfully by permutations of the finite chain $\mathrm{Im}(e) = e(Q) \subseteq Q_{\mathrm{FP}}$. The only order-preserving permutation of a finite chain is the identity, and there is at most one order-reversing one, the reversal, so $|G| \leq 2$. Thus, the only nontrivial group appearing is $\mathbb{Z}/2\mathbb{Z}$, whose instantiation was shown in Lemma J.5. $\square$

**Corollary J.7** (Floating-point cores have elementary abelian 2-group content). *Every diagonal SSM core over* FP*, with NaNs excluded, self-map updates, and arbitrary signed multipliers, has a transition monoid whose subgroups are elementary abelian 2-groups.*

*Proof.* By Lemma J.6, each coordinate monoid has only trivial or $\mathbb{Z}/2\mathbb{Z}$ subgroups. The core monoid embeds in $M_1 \times \cdots \times M_m$, so each of its subgroups embeds in a product of copies of $1$ and $\mathbb{Z}/2\mathbb{Z}$, hence is elementary abelian of exponent at most 2. $\square$

### J.2.2. UNSIGNED INTEGERS

**Lemma J.8** (Unsigned integers contain aperiodic monoids). *For $\mathfrak{M} = \mathrm{INT}^p$ the one-dimensional affine updates generate nontrivial aperiodic monoids.*

*Proof.* The update with $a = 2$, $b = 0$ is $F(q) = 2q \bmod 2^p$, which sends every state to 0 after $p$ iterations, so $F^p = F^{p+1}$ and $\langle F \rangle_{\mathrm{mon}}$ is aperiodic. $\square$

**Lemma J.9** (Unsigned integers contain cyclic 2-groups). *For $\mathfrak{M} = \mathrm{INT}^p$ and every $k \leq p$, the one-dimensional affine updates contain a cyclic group isomorphic to $\mathbb{Z}/2^k\mathbb{Z}$.*

*Proof.* The translation $F^{\mathrm{INT}^p}_{1,1}(q) = q + 1 \bmod 2^p$ is a single $2^p$-cycle on $\mathbb{Z}/2^p\mathbb{Z}$, hence a permutation of order $2^p$. For $k \leq p$ the step of size $2^{p-k}$ has order $2^k$, giving an explicit copy of $\mathbb{Z}/2^k\mathbb{Z}$. $\square$

**Corollary J.10** (Unsigned-integer cores have 2-group content). *Every diagonal SSM core over $\mathrm{INT}^p$ has a transition monoid whose subgroups are finite 2-groups, and a single core already realizes $\mathbb{Z}/2^k\mathbb{Z}$ for every $k \leq p$.*

*Proof.* The bijective affine self-maps of $\mathbb{Z}/2^p\mathbb{Z}$ are the maps $q \mapsto aq + b$ with $a$ a unit of $\mathbb{Z}/2^p\mathbb{Z}$, that is, with $a$ odd; they form the group $\mathrm{AGL}(1, \mathbb{Z}/2^p\mathbb{Z})$,[8] of order $2^{p-1} \cdot 2^p = 2^{2p-1}$, a 2-group. An affine self-map $q \mapsto aq + b$ is idempotent precisely when $a^2 = a$ and $ab = 0$ in $\mathbb{Z}/2^p\mathbb{Z}$. In $\mathbb{Z}/2^p\mathbb{Z}$, $a^2 = a$ forces $a \in \{0, 1\}$: either $a = 0$ (a constant map, with $b$

---

[8]The affine general group of degree 1 over the ring $\mathbb{Z}/2^p\mathbb{Z}$.

arbitrary), or $a = 1$ together with $b = 0$ (the identity). Let $G$ be a subgroup of a coordinate monoid, with identity idempotent $e$. If $e$ is a constant map, $G$ acts faithfully on the singleton $\mathrm{Im}(e)$ and is trivial; if $e = \mathrm{Id}$, then $G \leq \mathrm{AGL}(1, \mathbb{Z}/2^p\mathbb{Z})$ and is a 2-group. Every subgroup of a coordinate monoid is therefore a 2-group, and the same holds for the core monoid, which embeds in the product of coordinate monoids. Realizability of $\mathbb{Z}/2^k\mathbb{Z}$ is Lemma J.9. □

*Table 2.* Group content of a single diagonal SSM core, by recurrence and arithmetic model. Each row lists the subgroups realizable within a single core and the smallest pseudovariety that contains the core monoid.

| Recurrence | Model | Subgroups | Core monoid lies in |
|---|---|---|---|
| nonnegative | FP | only the trivial subgroup | $\mathbf{Ap}$ |
| signed | FP | elementary abelian 2-groups | $\mathbf{Ap} \vee \langle \mathbb{Z}/2\mathbb{Z} \rangle$ |
| nonnegative | $\mathrm{INT}^p$ | 2-groups, including $\mathbb{Z}/2^k\mathbb{Z}$ for $k \leq p$ | $\mathbf{Ap} \vee \mathbf{G}_2$ |

**Reading the table.** The two regimes, signed FP and nonnegative $\mathrm{INT}^p$, realize the same pseudovariety at the level of a single core, $\mathbf{Ap} \vee \langle \mathbb{Z}/2\mathbb{Z} \rangle$, even though their subgroup data differ. Krohn–Rhodes sees only the simple group divisors, and the only simple 2-group is $\mathbb{Z}/2\mathbb{Z}$. The richer subgroup structure of an $\mathrm{INT}^p$ core, namely cyclic 2-groups of any order up to $2^p$, is therefore absorbed by the wreath-product closure. The difference between the two regimes is a depth-for-width trade within the same pseudovariety, made explicit in Remark J.15. Here, $\vee$ denotes the join of pseudovarieties, i.e., $\mathbf{A} \vee \mathbf{B}$ is the smallest pseudovariety containing pseudovarieties $\mathbf{A}$ and $\mathbf{B}$.

### J.3. Upper bounds

The divisibility chain Eq. (3.1) turns each core-level fact above into a constraint on the recognized language. The accepting core of an SSM acceptor is memoryless: its state space is a singleton; hence, its transition monoid is trivial. By Corollary 3.21, the augmented wreath product therefore has the same group content as the base, and we omit the accepting core from the group-theoretic bookkeeping below.

**Theorem J.11** (Nonnegative floating point). *Let $\mathcal{A}$ be an SSM acceptor whose base is a nonnegative diagonal SSM over* FP *with fixed evaluation semantics. Then $M(\mathcal{L}(\mathcal{A}))$ is aperiodic.*

*Proof.* By Corollary J.4, every core monoid lies in $\mathbf{Ap}$. Since $\mathbf{Ap}$ is closed under wreath product, $W_{\mathcal{R}+} \in \mathbf{Ap}$. By Eq. (3.1), $M(\mathcal{L}(\mathcal{A})) \prec W_{\mathcal{R}+}$, and $\mathbf{Ap}$ is closed under division. □

**Example J.12.** *The parity language $\mathcal{L} = \{w \in \{0,1\}^* : w$ has an even number of 1s$\}$ has syntactic monoid $\mathbb{Z}/2\mathbb{Z} \notin \mathbf{Ap}$. By Theorem J.11, no nonnegative diagonal SSM over* FP *recognizes $\mathcal{L}$.*

Allowing the multiplier $-1$, as suggested by Sarrof et al. (2024), adds the factor $\mathbb{Z}/2\mathbb{Z}$ but nothing beyond it.

**Theorem J.13** (Signed floating point). *Let $\mathcal{A}$ be an SSM acceptor whose base is a diagonal SSM over* FP *with fixed evaluation semantics and arbitrary signed multipliers. Then every group divisor of $M(\mathcal{L}(\mathcal{A}))$ is isomorphic to $\mathbb{Z}/2\mathbb{Z}$; equivalently, $M(\mathcal{L}(\mathcal{A}))$ divides a wreath product of flip-flops and copies of $\mathbb{Z}/2\mathbb{Z}$.*

*Proof.* By Corollary J.7, every core monoid has only elementary abelian 2-group subgroups, and the same therefore holds for $W_{\mathcal{R}+}$ by closure under wreath product and division. A simple subgroup of an elementary abelian 2-group is trivial or $\mathbb{Z}/2\mathbb{Z}$, so the only nontrivial Krohn–Rhodes group factors (Theorem I.3) of $W_{\mathcal{R}+}$ are copies of $\mathbb{Z}/2\mathbb{Z}$, and $W_{\mathcal{R}+}$ divides a wreath product of flip-flops and copies of $\mathbb{Z}/2\mathbb{Z}$. The conclusion follows from $M(\mathcal{L}(\mathcal{A})) \prec W_{\mathcal{R}+}$ (Eq. (3.1)) and transitivity of division. □

**Theorem J.14** (Unsigned integers). *Let $\mathcal{A}$ be an SSM acceptor whose base is a diagonal SSM over* $\mathrm{INT}^p$. *Then every group divisor of $M(\mathcal{L}(\mathcal{A}))$ is a finite 2-group; equivalently, $M(\mathcal{L}(\mathcal{A}))$ divides a wreath product of flip-flops and copies of $\mathbb{Z}/2\mathbb{Z}$.*

*Proof.* By Corollary J.10, every core monoid has only finite 2-group subgroups, and the same therefore holds for $W_{\mathcal{R}+}$. A finite 2-group has $\mathbb{Z}/2\mathbb{Z}$ as its only simple subquotient (since every $p$-group has a nontrivial center), so the only nontrivial Krohn–Rhodes group factors of $W_{\mathcal{R}+}$ are copies of $\mathbb{Z}/2\mathbb{Z}$, and $W_{\mathcal{R}+}$ divides a wreath product of flip-flops and copies of $\mathbb{Z}/2\mathbb{Z}$. The conclusion follows from Eq. (3.1) and transitivity of division. □

*Remark* J.15 (Same languages, different depth). Theorems J.13 and J.14 characterize the *same* class of languages: those whose syntactic monoid has only 2-groups as group divisors. The two regimes differ at the level of a single core, not of the language. A signed FP core realizes only elementary abelian 2-groups (Corollary J.7), so it produces $\mathbb{Z}/4\mathbb{Z}$ only across two layers, through $\mathbb{Z}/4\mathbb{Z} \prec \mathbb{Z}/2\mathbb{Z} \wr \mathbb{Z}/2\mathbb{Z}$; an INT$^p$ core realizes $\mathbb{Z}/2^k\mathbb{Z}$ within a single layer (Lemma J.9). The difference is thus a depth-for-width trade, and the algebraic view exhibits it as one.

**Example J.16.** *The language* $\mathcal{L} = \{w \in \{0,1\}^* : \text{the number of } 1s \text{ is a multiple of } 3\}$ *has syntactic monoid* $\mathbb{Z}/3\mathbb{Z}$. *A group divisor of a monoid covered by Theorem J.13 or Theorem J.14 is a 2-group, and* $\mathbb{Z}/3\mathbb{Z}$ *is not, so no diagonal SSM over* FP *or over* INT$^p$ *recognizes* $\mathcal{L}$.

**Corollary J.17.** *For any integer $k$ that is not a power of $2$, no diagonal SSM over* FP *or over* INT$^p$ *recognizes the language of words whose number of $1$s is divisible by $k$.*

The same impossibility for FP follows, by a different route, from the eigenvalue analysis of Grazzi et al. (2025, Theorem 2).

## J.4. A lower bound

The upper bounds leave room for parity under unsigned integers, and an explicit construction realizes it with a nonnegative recurrence.

**Example J.18** (Parity over unsigned integers). *The parity language* $\mathcal{L} = \{w \in \{0,1\}^* : w \text{ has an even number of } 1s\}$ *is recognized by a nonnegative diagonal SSM over* INT$^p$.

*Proof.* The syntactic monoid of $\mathcal{L}$ is $\mathbb{Z}/2\mathbb{Z}$, realizable by a single core by Lemma J.9, so it remains to give an explicit acceptor. Take one layer, $N = 1$, with $m = d = 1$ and 4-bit unsigned integers, so $Q = X = Y = \mathbb{Z}/16\mathbb{Z}$. Encode $e(0) = 0$ and $e(1) = 1$, and set $A(0) = A(1) = 1$, $B(0) = 0$, $B(1) = 8$, so the recurrence is

$$q' = A(x)\,q + B(x)\,x = \begin{cases} q + 8 \pmod{16}, & x = 1, \\ q, & x = 0. \end{cases}$$

The transition matrix is nonnegative. Take initial state $q_0 = 15$, so a prefix with an even number of 1s yields state 15, and one with an odd number yields state 7. Set $C(0) = C(1) = 1$, so the readout equals the state, and let the accepting core read the state alone, with accepting region $\{15\}$. The empty word, having zero 1s, leaves the state at 15 and is accepted, in agreement with $\varepsilon \in \mathcal{L}$, and a word $w \in \Sigma^*$ is accepted exactly when its number of 1s is even. $\square$

Read against Sarrof et al. (2024, Theorem 2), which states that no nonnegative diagonal SSM in finite precision recognizes parity, Example J.18 shows the claim to depend on the unstated assumption that the arithmetic is floating point: it holds over FP by Theorem J.11 and fails over INT$^p$.

**Finite against unbounded precision.** An SSM, all of whose layers have a finite state space, recognizes only regular languages, since a finite wreath product of finite monoids is finite and every language recognized by a finite monoid is regular. Unbounded precision changes the regime: an unbounded integer data type allows the core state to grow with the input, inducing an infinite, length-indexed monoid action and placing the model beyond the finite-automaton boundary, where non-regular behavior becomes available.

