# OpenReview forum: "An Algebraic View of the Expressivity of Recurrent Language Models"
_ICML.cc/2026/Conference — ICML 2026 regular_

### Official Review · Reviewer_BFBz · 2026-03-10

**Soundness:** 4
**Presentation:** 2
**Significance:** 3
**Originality:** 4
**Overall Recommendation:** 4
**Confidence:** 4

**Summary:**

This paper begins with an overview that there are many contradictory (or hard to reconcile) theoretical results about the computational expressivity of RNNs and transformers (in terms of what formal languages they can recognize). A main source of these discrepancies is that in these analyses, they model arithmetic differently- e.g. as perfect over reals, or as floating point, or as the precision being a parameter, etc— and that this choice is consequential. This paper introduces a new formalism: they first define RNNs in a very generic / broad way, and then show how the RNN’s operation (as it takes in new tokens / updates its internal state and outputs) can be captured by viewing it as a “transformation semigroup”, i.e. a semigroup with n action. Roughly, the semigroup is that of functions/actions indexed by tokens (inputs) in the vocabulary, and the action is on the set of internal / hidden states of the model. They prove a kind of compositionally; an entire multilayer RNN is captured by its corresponding transformation semigroup which is in turn a kind of composition, namely the left “wreath product” of layer-wise transformation semigroups (where the action is on the space of states of the single hidden layer, semigroup elements are functions on these indexed by inputs to that layer). Next they explain how this formalism is further refined / restricted by specifying the model of arithmetic / evaluation of operations on the reals; fixing such a model restricts the semigroup to a “realizable” version. Next, they use their formalism to reproduce some lower bounds (impossibility results) and upper bounds (showing an RNN model can indeed recognize a specific language), namely for SSMs (state space models) with varying models of arithmetic evaluation.

**Compliance With Llm Reviewing Policy:**

Affirmed.

**Key Questions For Authors:**

1) As mentioned in strengths/weaknesses, the real “proof” for the need for this new formalism would be a new (previously unknown and to some degree surprising) upper/lower bound that can be relatively easily derived with the framework.

2) Also as mentioned in strengths/weaknesses, does a similar formalism exist for transformers (arguably more relevant currently, for language modeling, than RNNs?) perhaps there is a relatively simple “port” of this idea, and if so would it yield lower/upper bounds?

3) How do we know that the *algebraic* aspect of this formalism is so useful… for very specific languages, like even / mod n length strings, etc., it helps because those languages have specific symmetries that result in “syntactic monoids” that you can check for in the model’s semigroup (if I understood correctly). However even for a broader class of formal languages— like regular languages, CFGs, etc., which are of real interest in this area (can a specific architecture recognize them?) will this approach work? This is not made clear… it seems that it may not, and rather just the part of the formalism that captures memory / information bottlenecks etc. could be used to prove lower/upper bounds, but then one isn’t gaining much from the highly algebraic view.

Some MINOR points (not major questions):
-abstract reads well / clear to general audiences until “by separating universal algebraic structure…” - it’s fine but could be improved substantially / increase interest in the paper by being made less technical and comprehensible to general ML and/or CS audience
- Line 87, if you want to capitalize, capitalize both Platonic and Ideal
- Not sure “basic semigroup” theory is not familiar to general ML / CS audiences.. this is quite a unique paper in this regard, so I would suggest that definitions need to be presented more thoroughly / rigorously. For instance definition 2.1 introduces “transformation semigroup” and then immediately talks about a “semigroup S” without the “X” part — this Is confusing. A homomorphism from a language (subset of strings) to the semigroup is also undefined - what is the algebraic structure on a set of strings? Why is there a “unique minimal such S”?
- 108, reversed quotation marks on “lower”
- 92 right side; “in the way it they are…” -> “in the way they are”
- What is a total map in the context of line 160? Not defined… as this is the only thing differentiating “algebraic RNN core” form “RNN core”. Note that in TCS total map means onto.
- 191 quotation mark fix
- 259, right side — you have flipped the meaning of upper ground / lower bound, as used in computer science. Upper bounds = existence theorems, an *upper* bound on the complexity of a problem (no harder than this, because “this” solves it), lower bound = impossibility (it is *at least* as hard as this, because “this” doesn’t solve it)
- Also 259, (3) introduces lots of new jargon that is hard to understand — I suggest to either define, or somehow simplify presentation for readability (e.g. “cascade” is used for the first time, “up to the chosen wiring/observability” is also abstruse)
- 319 right side (“what semigroups can appear?”) where did “T” come from? is it the same as F^A_u (why introduce new notation?)
- 7.3 title - reminder that impossibility = lower bounds

**Limitations:**

Yes

**Strengths And Weaknesses:**

The paper is in a very interesting area — formal interpretability / expressivity of machine learning models, specifically recurrent neural networks. Formal, mathematical, algebraic approaches to understanding what such models can and cannot do are very interesting and needed, especially in an environment where empirical / informal claims and investigations greatly outnumbers theory.

Specifically trying to capture RNN computation as an algebraic object is especially challenging and original, and the formalism introduced is mostly comprehensible and somewhat elegant. The fact that they are able to reproduce, more simply / from the same formalism, several formal-language computability results is evidence that their new formalism is interesting. These are the paper’s greatest strengths. It could be improved first and foremost by making the presentation a bit more digestible / comprehensible / self-contained; since ICLM is a general machine learning conference, one can assume prior knowledge in learning, model architectures to some extent, even some pure TCS and language theory, but perhaps not semigroup algebra; while the page limit / density of the information makes this challenging, perhaps there are a few easy ways to improve comprehensibility for general ML audiences (especially if the authors wish for this framework / algebraic approach to become well-known / highly studied). Second, as mentioned the paper does demonstrate some of what can be done with this formalism, but it is still overall in the baby stages — while quite ambitious, if they could show a new lower/upper bound that can be derived with the formalism, or if they could suggest a possible direction like this, it would draw more interest to this paper. Or, if this formalism could yield bounds for more complicated formal language classes (like CFGs, which are of interest in the study of “what formal languages models recognize”), rather than just very simple languages with symmetries. Relatedly, the introduction mentions transformers and some formal results concerning them, but the paper is dedicated to RNNs, so a suggestion / sketch / indication of how a similar formalism could work for transformers, and whether/how it would yield upper/lower bounds for that architecture, could be another way to increase the appeal of the paper to the general ML community.

---

> ### Author Rebuttal · Authors · 2026-03-30
>
> Thank you for acknowledging the originality and elegance of our framework, and for taking the time to carefully read our paper and provide detailed feedback.
>
> ### Responses to General Points
>
> > It could be improved first and foremost by making the presentation a bit more digestible…
>
> We agree and will revise the paper with a general ML audience more explicitly in mind, since semigroup algebra cannot be assumed as standard background at ICML.
>
> > Second, as mentioned the paper does demonstrate some of what can be done…
>
> The presented examples demonstrate the mechanics of the framework, not its scope. The formalism is not intrinsically tied to finite monoids, but concrete applications beyond the current regular-language examples are left for future work, which we will state explicitly in revision.
>
> ### Key Questions
>
> > As mentioned in strengths/weaknesses, the real “proof” for the need for this new formalism would be a new (previously unknown and to some degree surprising) upper/lower bound…
>
> We agree that the strongest evidence for a new formalism is that it yields genuinely new or unexpectedly transparent expressivity statements. A simple example of a surprising fact resulting from a focus on numerical semantics is that under symmetric saturation semantics (e.g., saturated fixed-point arithmetic) with fixed left-to-right evaluation, an RNN with a nominally linear state update can realize a recurrence with ReLU nonlinearity.
>
> *LRNN simulating ReLU-RNN:* If the representable range is $[-C, D]$ with saturating addition/subtraction, then
> $$
> z \mapsto ((z - C) + C)
> $$
> is exactly $\max(z, 0)$, since for $z \leq 0$, the first subtraction saturates to $-C$ and the second addition returns 0, while for $z > 0$ no saturation occurs and the value is preserved. Consequently, an LRNN with update
> $$
> x_{t+1} = ((A x_t + B u_t) - C) + C
> $$
> computes exactly
> $$
> x_{t+1} = \mathrm{ReLU}(A x_t + B u_t)
> $$
> under these numerical semantics, even though the update is syntactically linear in the underlying arithmetic operations.
>
> This has immediate expressivity consequences: ReLU Elman RNNs can simulate arbitrary DFAs (via the Minsky construction, see [Korsky and Berwik (1996)](https://arxiv.org/abs/1906.06349) or [Svete and Cotterell (2023)](https://arxiv.org/abs/2310.05161), among others), whereas RNNs with purely linear recurrent updates cannot, under non-saturated semantics ([Merrill et al., 2024](https://arxiv.org/abs/2404.08819), Theorem 4.2), so changing only the numerical semantics can move the same nominal architecture between fundamentally different language-theoretic regimes.
>
> This illustrates the core motivation for our framework: meaningful expressivity bounds must be stated relative to the induced transition structure under fixed operational semantics, not at the level of the symbolic recurrence equation alone.
>
> > Also as mentioned in strengths/weaknesses, does a similar formalism exist for transformers…
>
> *Transformer extension:* We are developing a semigroup-theoretic formalization of autoregressive multilayer language models, which is intended to also extend to transformers. Although attention makes the extension more subtle than in the RNN case, because updates depend on information propagated from the entire prefix of hidden states, we have identified a suitable notion of transformer state under which the computation can still be analyzed as a causal cascade of layer transitions. In finite precision, this again induces finite transition semigroups, with layers composing in a wreath-product-like fashion and tokens acting as cascade inputs, opening the door to Krohn–Rhodes-style expressivity analysis.
>
> > How do we know that the algebraic aspect of this formalism is so useful…
>
> The algebraic component is most useful when the target language imposes structural constraints - modularity, symmetry - that must appear in the model's transition algebra. In the finite-state setting, as Krohn-Rhodes theory shows, every regular language monoid decomposes into prime components of group-like or acyclic nature. However, hierarchical decomposition is not conceptually restricted to finite monoids: for instance, unbounded counting behavior can be viewed in terms of infinite algebraic components (e.g., the bicyclic monoid in Dyck-like settings). Due to our focus on fixed-parameter and fixed-precision settings in our case study, our current examples demonstrate the method only in regular-language settings, and we will make this scope explicit.
>
> ### Other Comments and Suggestions
> > you have flipped the meaning of upper ground / lower bound...
>
> In expressivity analysis, the convention is standard: an upper bound limits what a model class can express (impossibility), and a lower bound provides a construction that witnesses expressivity (existence). This is the dual of complexity-theoretic convention because expressivity is dual to hardness.
>
> We agree with the other minor points raised by the reviewer and will fix them in revision.

---

> > ### Author Rebuttal · Reviewer_BFBz · 2026-04-03
> >
> > Thank you for your detailed rebuttal! If I understood correctly, the insight that ReLU Elman RNNs can simulate arbitrary DFAs is a new one? If that is the case it can/should be added to this paper and the novelty emphasized, as stemming / having simple proof in your introduced framework! Please let me know if this is the case.
> >
> > Also please do try to revise, to the extent possible, to make the presentation a bit easier / more accessible. For example moving some proofs / aspects to the appendix, simplifying / giving more intuitive summaries in earlier sections, etc. I see this is a concern also raised by other reviewers (I assume you will make this revision!) I don't know if you have any examples of what you have rewritten or could rewrite to make this more accessible / interesting to the ICML audience?

---

> > > ### Author Response · Authors · 2026-04-07
> > >
> > > To clarify: it was already known that bounded finite-precision ReLU Elman recurrences can simulate arbitrary deterministic DFAs; that fact itself is not new.
> > >
> > > Our contribution is different. We show that, under explicitly specified staged clamped finite-precision semantics, even a nominally linear recurrence can realize the same thresholding behavior as a ReLU recurrence, and therefore simulate arbitrary deterministic DFAs as well. Thus, the novelty is not a new DFA simulation theorem for ReLU Elman recurrences, but a new expressivity result for nominally linear recurrences under realistic arithmetic semantics.
> > >
> > > This directly illustrates our main thesis that arithmetic semantics is part of the model semantics. We agree that making this example explicit will substantially improve accessibility for an ML audience, and in the revision we will add it to the main text as an intuitive worked example, with the more technical details moved to the appendix.
> > >
> > > ## Example 5.3 (Clamping can realize ReLU inside a nominally linear recurrence)
> > >
> > > The written recurrence alone does not determine the implemented transition map. In particular, a source-level affine update may become nonlinear once finite-precision semantics are fixed. This is not merely a pathological possibility: in standard fixed-point and integer-only implementations, matrix products are typically accumulated in a wider register and then requantized or saturated when cast back to the target format, so clamping already appears as part of realistic execution semantics (Jacob et al., 2018; Kim et al., 2021). The crucial issue is therefore not whether clamping occurs, but where the clamp barriers occur in the staged implementation.
> > >
> > > To make this explicit, compare the usual Elman recurrence
> > > $$
> > > x_t=\text{ReLU}(A_0 x_{t-1}+B_0 u_t+b_0)
> > > $$
> > > with the source-level affine update
> > > $$
> > > x_t = A x_{t-1}+B u_t+b_1+b_2+b_3.
> > > $$
> > > Syntactically, the latter contains no nonlinearity. Now fix a consistent evaluation order and a staged clamped signed-integer semantics with coordinatewise saturation
> > > $$
> > > \text{clamp}(z)=\max(\min(z,L),-L).
> > > $$
> > > Assume that the affine term $A x_{t-1}+Bu_t+b_1$ is first materialized in the target fixed integer format, and that the subsequent bias additions are then each performed in that same clamped format. The implemented update is therefore
> > > $$
> > > x_t=\text{clamp}\Big(\text{clamp}\big(\text{clamp}(A x_{t-1}+Bu_t+b_1)+b_2\big)+b_3\Big).
> > > $$
> > > The intermediate clamp barriers are part of the semantics.
> > >
> > > ### Proposition 5.4
> > >
> > > Let
> > > $$
> > > x_t=\text{ReLU}(A_0 x_{t-1}+B_0 u_t+b_0).
> > > $$
> > > Choose
> > > $$
> > > A=A_0,\qquad B=B_0,\qquad b_1=b_0,\qquad b_2=-L\mathbf{1},\qquad b_3=L\mathbf{1}.
> > > $$
> > > Let
> > > $$
> > > v=A_0 x_{t-1}+B_0 u_t+b_0.
> > > $$
> > > If $v\in[-L,L]^N$, then the staged clamped affine recurrence computes
> > > $$
> > > x_t=\text{ReLU}(v)
> > > $$
> > > exactly, coordinatewise.
> > >
> > > ### Proof
> > >
> > > Fix a coordinate $i$. Since $v_i\in[-L,L]$, the first clamp is inactive, so $\text{clamp}(v_i)=v_i$. At the second stage,
> > > $$
> > > \text{clamp}(v_i-L)=
> > > \begin{cases}
> > > -L & \text{if } v_i\le 0,\\\\
> > > v_i-L & \text{if } v_i>0,
> > > \end{cases}
> > > $$
> > > because $v_i-L\le -L$ in the first case and $v_i-L\in(-L,0]$ in the second. After adding $L$ and clamping once more, we obtain
> > > $$
> > > \text{clamp}(\text{clamp}(v_i-L)+L)=
> > > \begin{cases}
> > > 0 & \text{if } v_i\le 0,\\\\
> > > v_i & \text{if } v_i>0.
> > > \end{cases}
> > > $$
> > > Hence the output is exactly $\max(v_i,0)=\text{ReLU}(v_i)$. $\square$
> > >
> > > Intuitively, the gadget “subtract $L$, clamp, add $L$” turns the lower saturation boundary into a zeroing threshold: negative values are pushed to $-L$ and return as $0$, while positive values remain in the unsaturated regime and survive unchanged.
> > >
> > > ### Corollary 5.5
> > >
> > > Any DFA simulation theorem for bounded finite-precision ReLU Elman recurrences transfers to this staged clamped affine model.
> > >
> > > ### Proof
> > >
> > > Classical threshold-network constructions, going back to Minsky, show how finite-state dynamics can be embedded into recurrent networks via thresholding behavior. For the bounded finite-precision setting relevant here, Korsky and Berwick (2019) show constructively that finite-precision one-hidden-layer ReLU recurrences are exactly as computationally powerful as deterministic finite automata, using only integers in a bounded range. By Proposition 5.4, each such ReLU update can be replaced, within the same bounded working region, by the staged clamped affine update above. Therefore, the same hidden-state transition system, and hence the same DFA simulation, is realized. $\square$
> > >
> > > This does not contradict prior limitations for linear recurrences (e.g., Merrill, 2024). Those results assume computational semantics under which the recurrence remains genuinely linear. By contrast, we show that whether a recurrence is genuinely linear cannot be read off from the written formula alone. Once finite-precision clamping is part of the operational semantics, a nominally affine recurrence may already contain the thresholding nonlinearity on which classical automata simulations rely.

---

### Official Review · Reviewer_tEYy · 2026-03-12

**Soundness:** 4
**Presentation:** 3
**Significance:** 4
**Originality:** 4
**Overall Recommendation:** 5
**Confidence:** 4

**Summary:**

This paper addresses a fundamental crisis in the theoretical analysis of recurrent neural networks (RNNs, LSTMs, SSMs): the literature is riddled with seemingly contradictory expressivity claims—ranging from strictly finite-state to fully Turing complete. The authors compellingly argue that this tension stems from sloppy numerical semantics (imprecise use of terms like finite precision). Traditional machine learning theory analyzes the "Platonic Ideal" of networks operating over continuous real numbers (R), but physical models operate on finite-precision hardware (e.g., floating-point or quantized integers). Because discretization breaks an algebraic laws as fundamental as  associativity, the theoretical guarantees derived from these are not reliable. To resolve this, the authors propose a rigorous, unifying algebraic framework that treats discretization not as a negligible approximation error, but as a core structural property of the model. By fixing the arithmetic semantics and evaluation order, treating layers as semigroups, depth as a "realized" wreath product, the authors show a recipe of how to prove expressivity bounds with their framework. They validate their framework through a case study on non negative Diagonal State Space Models (SSMs), where they prove the same results as before (Sarrof et al 2024), and at the same time, show how with a different set of choices with regards to the numerical semantics, they can prove a contradictory results, highlighting how essential it is to not treat things like finite precision so lightly, and at the same time offers a new paradigm that is still elegant enough to do so for recurrent language models.

**Compliance With Llm Reviewing Policy:**

Affirmed.

**Final Justification:**

The rebuttal process (reading others reviews and the authors comments) helped reinforce my prior assessment. I think that the paper raises an important question, gives a way to answer it, shows how a simple choice flips previously established results and also at the same time then allows for new expressivity results to be established. These seem important to me, and I would like this paper to be accepted to bring to light these issues and this kind of a solution more.

**Key Questions For Authors:**

There are minor LaTeX issues that the authors can try and fix, things like quotation marks, using '' instead of `` to start quotes, but that's not critical. My main questions for the authors are as follows --

- Significance of Mealy Machines (Line 162 / Footnote 2): You explicitly note that when the sets X, U, and Y are finite, the algebraic RNN core corresponds to a Mealy machine. Why was this mentioned ? What is the significance of this?

- Extension to Transformers: The paper focuses specifically on recurrent architectures (RNNs, LSTMs, SSMs). Could the authors speculate whether the current framework can somehow easily be extended to study transformers as well or whether a similar framework for the same would be a lot more tedious ?

**Limitations:**

The authors can try and include a limitations section somewhere in their paper.

**Strengths And Weaknesses:**

**Strengths**

*Elegant and Actionable Framework*: The authors do not just point out a problem; they provide a comprehensive solution. Translating RNN layers into transformation semigroups and architectural depth into wreath products is an elegant use of abstract algebra. Furthermore, providing the 4-step "Meta-Paradigm" gives the ML community a highly actionable recipe for standardizing future expressivity proofs.

*Compelling Case study on Diagonal SSMs*: The case study on Diagonal SSMs grounds the heavy abstraction perfectly. By using their framework to formally re-prove how non-negative constraints or floating-point rounding limit an SSM's ability to count, and in addition showing how a change in the choice of their numerical semantics, they can get a contradictory result of the same statement, the authors highlight the value of their method.

*A Much-Needed Reality Check for ML Theory*: This paper also serves as a mathematically rigorous position paper. The authors draw attention to critical blind spots in the literature. For instance, pointing out how discretization destroys associativity—and therefore breaks standard proofs.

**Weakness**

*High Barrier to entry*: The primary weakness of this paper is its accessibility. The framework heavily relies on abstract algebra (transformation semigroups, magmas, wreath products, Krohn-Rhodes theory), which a typical machine learning researcher may or may not be aware of. While the math is rigorous, the sheer density of the notation might prevent the paper from having the widespread impact it deserves. Maybe having examples/visual aids alongside every definition (in the appendix or otherwise) could be really helpful.

*Limited to expressivity*: The framework is strictly an analysis of the forward pass and expressivity, and not about learnability. A finite-state network might theoretically be able to recognize parity under wraparound integer semantics, but can gradient descent actually find those weights, such a framework can't do that. Perhaps adding a discussion line about the extent of the utility of the current framework would be useful.

---

> ### Author Rebuttal · Authors · 2026-03-30
>
> Thank you very much for your positive review and for recognizing the value of a comprehensive framework and step-by-step recipe for rigorous expressivity analysis that highlights and avoids common blind spots in the literature.
>
> ### Responses to Specific Points
>
> > High Barrier to entry
>
> This is very helpful feedback. We agree that the current presentation poses a high barrier to entry, especially for readers without prior exposure to semigroup theory or wreath products. The formalism necessarily relies on fairly abstract algebraic machinery, but the paper should do more to make the framework more navigable and intuitive. In revision, we will work to improve accessibility by pairing the main formal definitions with concrete examples and visual aids, and by adding more explanatory text to guide the reader through the purpose of the main constructions rather than introducing them only in their most abstract form.
>
> > Limited to expressivity
>
> We agree. Our framework concerns expressivity, not learnability: it characterizes what the forward computation can represent under fixed architectural and arithmetic assumptions, but does not address whether optimization will find those representations. We will make this limitation explicit. We nevertheless believe the analysis is useful because it separates optimization issues from genuine architectural constraints: impossibility results rule out behaviors regardless of training, whereas constructive results identify mechanisms that are, in principle, sufficient. This type of result has already had an impact in the SSM literature, where expressivity limitations for certain diagonal SSMs ([Sarrof et al., 2024](https://arxiv.org/abs/2405.17394)) helped motivate more expressive variants such as AUSSM ([Karuvally et al., 2025](https://arxiv.org/abs/2507.05238)) and DeltaProduct ([Siems et al., 2025](https://arxiv.org/abs/2502.10297)).
>
> ### Key Questions
>
> 1. The reason for mentioning Mealy machines is to connect our abstraction to a standard and familiar automata-theoretic object. This makes explicit that, under finite state spaces, our algebraic RNN core reduces to finite-state machines/transducers, which is precisely what allows us to connect the framework to existing work on finite semigroups and Krohn–Rhodes theory. We will clarify this in the revision.
>
> 2. This is a very relevant question. This paper is a first step in developing a semigroup-theoretic formalization of autoregressive multilayer language models that can, in principle, also be extended to transformers. See our response to reviewer BFBz for more details.

---

> > ### Author Rebuttal · Reviewer_tEYy · 2026-03-31
> >
> > I was already happy with the draft. The authors response are clear, and my concerns have been met. I would recommend the authors to keep their word and try and improve readability of the paper for the camera ready version. I will keep my score. I have also read the other reviews and the authors responses. The review of yne3 seems especially harsh to me, but it is understandable given the confidence score, and this speaks back to the difficulty of understanding the paper submitted by the authors. This makes improving the readability all the more important. I hope the author reviewer discussion bears fruit and clarifications can be made in both directions.

---

### Official Review · Reviewer_yne3 · 2026-03-15

**Soundness:** 1
**Presentation:** 2
**Significance:** 2
**Originality:** 2
**Overall Recommendation:** 2
**Confidence:** 2

**Summary:**

Recent works in language model reasoning attempt to characterize which algorithms such models can implement or which formal languages they can recognize (e.g. studying recurrent models via reductions to finite state automata and regular languages). The authors claim many of such results are not well-posed and that many proofs are highly architecture-specific and difficult to transfer across related models. The authors thus suggest a unifying algebraic framework for a class of recurrent models based on wreath products of transformation groups. They provide a single specific case study on diagonal SSMs. The authors suggest that their paradigm could unify and correct expressivity claims in the literature.

**Compliance With Llm Reviewing Policy:**

Affirmed.

**Final Justification:**

I have read the author responses to the other reviewers. In particular, I appreciate the new result provided by the authors in their response to reviewer BFBz in trying to extend the applicability. However, like reviewer BFBz, I do not fully understand the details or significance of this result, tying into a broader problem of the paper being overly technical and difficult to understand as reviewer BFBz mentioned. While this new result potentially addresses one aspect of my concern (which is difficult to assess), I still lean towards reject and believe a re-write of the paper would lead to a more conducive reviewing process.

**Key Questions For Authors:**

- Can you explain how this generalizes to models other than RNNS, e.g. Transformers (most common architectures for language today), etc.?
- Why is the wreath product the natural or chosen algebraic abstraction?
- Can you expand on the categorical perspective?

**Limitations:**

There is only a single mathematical case study on diagonal SSMs. Can you elaborate on what you think would happen in other cases? Moreover, I suggest including in the abstract the specific case study that the authors conducted, in order to make the contribution more explicit.

**Strengths And Weaknesses:**

**Strength**
- Case study on diagonal state space models helps concretize the framework, but its broader applicability still remains to be validated.

**Weakness**
- Soundness: analysis on more diverse models than just diagonal SSMs is lacking.
- Originality: A more in-depth related works section comparing and contrasting prior and similar works would help assess the originality better.
- Significance: The idea that there could be a paradigm that addresses issues found across the literature due to incorrect assumptions would hold significance, however the feasibility of the paradigm the authors propose still remains to be validated on more case studies.
- Section 8 discusses the categorical perspective, but there is a missing citation: "Categorical Deep Learning is an Algebraic Theory of All Architectures" Gavranović et al. (ICML 2024).
- No discussion of limitations or future work, no conclusion.

---

> ### Author Rebuttal · Authors · 2026-03-30
>
> Thank you for reviewing our paper and for acknowledging the applicability of our framework.
>
> ### Responses to Specific Points
>
> > Soundness: analysis on more diverse models than just diagonal SSMs is lacking.
>
> The diagonal SSM case study illustrates how the framework works, not a standalone collection of new separation results.
> The framework is directly applicable to Elman RNNs, LSTMs, and other recurrent architectures.
> All that is required is to specify the appropriate transition and wiring functions under the chosen numerical semantics.
>
> > Originality: A more in-depth related works section comparing and contrasting prior and similar works would help assess the originality better.
>
> We agree and will expand the related work discussion in revision. See our response to reviewer KwQy for a comment on how our treatment relates to prior bounded-precision work.
>
> > No discussion of limitations or future work, no conclusion.
>
> Will be added in revision.
>
> > There is only a single mathematical case study on diagonal SSMs. Can you elaborate on what you think would happen in other cases? Moreover, I suggest including in the abstract the specific case study...
>
> We agree that the abstract should name the case study explicitly. We chose diagonal SSMs for this because their expressivity has been a recent focus of study, e.g., by [Sarrof et al. (2024)](https://arxiv.org/abs/2405.17394), providing concrete claims to verify and critique. The diagonal case is not special in kind — once numerical semantics are fixed, our framework is applicable to any recurrent architecture. See our response to reviewer BFBz for a concrete example (saturation semantics inducing ReLU-like behavior in nominally linear recurrences).
>
>
> ### Key Questions
>
> 1. This paper is a first step in developing a semigroup-theoretic formalization of autoregressive multilayer language models that, in principle, also extends to transformers. See our response to reviewer BFBz for more details.
> 2. Wreath products are a natural and useful abstraction because multilayer recurrent computation is inherently a cascade: each layer carries its own state dynamics, while lower layers determine which updates act on higher layers. Once numerical semantics are fixed, each layer induces a transition semigroup, and the semigroup of the full network is obtained by hierarchical composition of these controlled actions. The wreath product is the canonical algebraic formalization of exactly such cascaded sequential systems. It is therefore the natural abstraction for deep RNNs.
> The relevance to expressivity is that formal language recognition is governed by the algebraic structure of the induced transition semigroup. In algebraic automata theory, wreath products are the classical operation for building complex transition semigroups from simpler ones, and in the finite-precision case, this lets us place multilayer RNNs directly in the setting of regular language recognition and Krohn–Rhodes-style analysis. At the same time, our formulation is not restricted to the finite-state regime: the same wreath-product perspective still describes hierarchical composition under non-finite-precision numerical semantics, where a purely automata-based account no longer applies. Thus, wreath products provide a uniform algebraic abstraction for expressivity both within and beyond the classical automata-theoretic setting.
> 3. We have read Gavranović et al. carefully and can readily see why the reviewer sensed a connection: both papers use algebraic language to study neural architectures, and both mention RNNs and automata. However, "algebraic" here refers to different mathematical traditions. Our work belongs to the Eilenberg tradition of algebraic automata theory (transformation semigroups, wreath product, Krohn-Rhodes), while Gavranović et al. operate in the categorical settings of monads, endofunctor (co)algebras, and the 2-category of parametric maps. More importantly, the questions differ: we seek a unified categorical language for describing the algebraic structure of the induced transition semigroups, and consequently, which expressivity claims are well-posed. Their framework presupposes the very associativity and functoriality that discretization destroys, so the problem central to our paper cannot even be stated within it.
> Note the submitted draft loosely called $\mathcal{F} : \mathbb{R}\text{-RNN} \to \text{Fin-RNN}$ a "functor", but since rounding breaks associativity and distributivity, $\mathcal{F}$ does not preserve morphisms. In revision, the section will be retitled "A Structural Perspective", $\mathcal{F}$ will be called a "discretization mapping", and the failure of functoriality will be stated explicitly.
> We are concerned that citing Gavranović et al. without substantive engagement could mislead readers into expecting shared technical content; we will add a footnote in the revised section 8 clarifying the distinction between the two frameworks.

---

> > ### Author Rebuttal · Reviewer_yne3 · 2026-04-04
> >
> > I thank the authors for their response. Upon consideration of other reviewers opinions and revisiting the paper I have decided to maintain my score because of the lack of examples demonstrating the feasibility of their framework across more diverse settings. Additionally, the rebuttal doesn't address the limitations (this being a single case study) or the lack of a clear path for future work to build on this.

---

> > > ### Author Response · Authors · 2026-04-04
> > >
> > > We respectfully disagree with the assessment that our rebuttal leaves core concerns unresolved.
> > >
> > > **Diverse settings.** Our framework applies as written to all recurrent layered sequence models (SSMs, RNNs, LSTMs, RWKV, etc.). The algebraic machinery is parameterized by the recurrence structure, not by the specific architecture. We would appreciate clarification on what additional diversity of settings the reviewer has in mind.
> > >
> > > **Lack of examples.** As detailed in our response to BFBz, we add a concrete example of linear recurrence simulating nonlinear recurrence in RNNs, directly demonstrating the framework's analytic use beyond the diagonal SSM case.
> > >
> > > **Single case study / limitations.** The diagonal SSM analysis is illustrative, not itself part of the claimed contribution. Applying the framework to other architectures follows the same procedure. We have committed to making this explicit in the revised draft, including a dedicated discussion of limitations.
> > >
> > > **Future work.** Our response to BFBz outlines two concrete directions: (1) extension to non-finite monoids (e.g., bicyclic monoids for Dyck languages and counting subtasks, requiring growing network size or precision), and (2) adaptation of the framework to transformers. Both will be incorporated into the revision.
> > >
> > >
> > > We have committed to integrating all of the above into the revised manuscript. Given these clarifications, we would welcome the reviewer specifying which concrete concerns remain that would warrant maintaining the current score.

---

### Official Review · Reviewer_KwQy · 2026-03-16

**Soundness:** 2
**Presentation:** 2
**Significance:** 2
**Originality:** 3
**Overall Recommendation:** 4
**Confidence:** 4

**Summary:**

This paper proposes an algebraic framework for analyzing the expressivity of recurrent neural architectures under explicitly specified arithmetic semantics. The main claim is that many existing expressivity results rely on unrealistic assumptions (e.g., real arithmetic with infinite precision), and that meaningful analysis must instead treat finite-precision arithmetic and evaluation order as intrinsic components of the computational model. The authors formalize recurrent architectures via transformation semigroups and wreath products, and illustrate the framework through case studies on diagonal state-space models under different arithmetic regimes. The paper is mathematically motivated and aims to bring tools from algebraic automata theory into modern sequence modeling. However, the technical contributions remain largely structural and upper-bound oriented, without engaging deeply with constructive automata equivalence results that already exist in the literature.

**Compliance With Llm Reviewing Policy:**

Affirmed.

**Final Justification:**

Have already addressed in ack section.

**Key Questions For Authors:**

Please read Weakness

**Limitations:**

Please read Weakness

======
Changed after rebuttal

**Strengths And Weaknesses:**

## Strengths

1. The most valuable aspect of the paper is the insistence that **computational semantics must be fixed before making expressivity claims**. The observation that floating-point arithmetic breaks associativity and therefore changes the algebraic structure of recurrent updates is correct and important. This point is often ignored in neural expressivity discussions, and the paper provides a mathematically coherent language for addressing it.

2. The use of transformation semigroups and wreath products is also well-motivated. From a formal language theory perspective, this is a natural way to reason about layered transition systems. The distinction between ambient and realized wreath products is conceptually meaningful and reflects the fact that internal state transitions in deep networks are constrained by architectural wiring rather than arbitrary inputs. This part of the work feels technically grounded and aligns with classical Krohn–Rhodes style reasoning.

3. The diagonal SSM case study is mathematically clean. It illustrates how different arithmetic models induce different group structures in the associated transition semigroup, and how this in turn constrains the class of recognizable modular languages. This example demonstrates that the proposed algebraic framework can yield nontrivial expressivity bounds.

## Weaknesses

The most significant issue is **missing engagement with prior bounded-precision neural memory literature**, particularly the NNTM / neural stack line of work by Stogin et al 2019 https://www.sciencedirect.com/science/article/abs/pii/S0020025523016201 and also Omlin and giles with 2nd order RNN (https://clgiles.ist.psu.edu/papers/JACM-1996-stable-encoding-rnn-automata.pdf). That line provides constructive simulations of DFA, PDA, and TM-level computation under bounded-precision neural dynamics and includes explicit stability results for neural memory operators. These results directly intersect with the central thesis of this paper, namely that expressivity must be analyzed under realistic arithmetic semantics.

My main concern is that the paper’s positioning is stronger than what is actually established. The central claim is that expressivity results for recurrent language models become misleading unless arithmetic semantics are fixed explicitly, and that the proposed algebraic framework provides the right way to do this. I agree with the general point that evaluation order, overflow, and rounding can change the computational object being studied. However, the novelty claim still feels overstated. There is already prior theoretical work on bounded-precision neural computation and finite-precision expressivity, so the paper should be much clearer about what is genuinely new here: is it the specific semigroup formalization, the realized wreath-product viewpoint, the diagonal SSM case study, or the broader methodological stance? As written, these are not separated cleanly enough.

From a mathematical perspective, the framework is much stronger on **upper bounds** than on actual characterization. The main technical results show that once a layer-wise transition algebra is identified, the global behavior is contained in an ambient or realized wreath product. That is a legitimate algebraic statement, but it is still only a structural containment result. It does not by itself tell us whether the bound is tight, whether the resulting semigroup is minimal, or whether the model can realize a corresponding language class in a robust sense. In automata theory, upper-bound machinery is only one half of an expressivity theory. Without matching lower bounds or constructive realizability results, the framework remains more classificatory than explanatory.

I also think the paper moves too quickly from arithmetic sensitivity to broad conclusions about practical expressivity. Finite precision certainly changes the induced transition system, but the conclusion that practical expressivity effectively collapses to finite-state behavior is too sweeping in the present form. That inference only goes through under additional assumptions: fixed state dimension, fixed precision, fixed evaluation order, no growing auxiliary storage, and a closed transition system. Without making those assumptions explicit, the statement is mathematically too coarse. In particular, finite precision does not automatically mean that every relevant recurrent model should be understood as “just” a finite automaton in the strongest formal sense; often the right object is a family of transition systems indexed by architecture size, sequence length, or precision level.

A related issue is that the paper does not fully separate **semantic correctness** from **computational significance**. Showing that two arithmetic regimes induce different semigroups is interesting, but the conceptual payoff depends on whether those differences translate into meaningful differences in recognized language classes or sequence-processing capabilities. In the current draft, that bridge is only partially built. The diagonal SSM examples are mathematically neat, especially for modular counting, but they are also quite specialized. They do not yet convince me that the proposed framework yields a broadly useful theory of recurrent language models beyond a small class of analytically convenient examples.

Finally, the scope of the concrete results is narrow relative to the ambition of the paper. Most of the technical payoff comes from diagonal SSM-style examples and modular language separations. That is a reasonable starting point, but the paper makes broader claims about recurrent language models as a class. To support that scope, I would expect at least one additional case study where the algebraic framework reveals something genuinely non-obvious about a less restricted recurrent architecture. Right now, the paper feels more like a promising algebraic program than a mature expressivity theory.

## Suggestions for improvement

1. Add a detailed comparison with NNTM / neural stack work, explicitly explaining how the proposed algebraic framework relates to bounded-precision simulations of DFA, PDA, and TM computation. Also have comparsion against this recent work "https://arxiv.org/html/2405.19222v2"

3. Clarify which results are new expressivity theorems versus reinterpretations of known algebraic facts.

4. Moderate claims about practical expressivity until supported by stronger formal analysis.

5. Develop the lower-bound / constructive side of the theory to complement the current upper-bound machinery.

---

> ### Author Rebuttal · Authors · 2026-03-30
>
> Thank you for your in-depth review and constructive feedback on our paper and for appreciating the value of a rigorous focus on numerical semantics when evaluating the expressivity of language models. Indeed, we believe that semigroup theory is the broadest and most natural way to reason about the expressivity of layered sequential systems and expose the numerical assumptions that make such analyses possible in the first place.
>
> ### Responses to Specific Points
> > The most significant issue is missing engagement with prior bounded-precision neural memory literature, particularly the NNTM / neural stack line of work by Stogin et al 2019...
>
> We thank the reviewer for pointing us to this line of work and will cite it in the revised version. However, these analyses are carried out in a continuous real-valued setting and do not specify evaluation order, rounding mode, overflow behavior, or the resulting failure of associativity in floating-point computation. Parts of their proofs rely on exact algebraic rewritings that are harmless over the reals but not under non-associative finite-precision semantics - precisely the distinction our paper isolates: “bounded precision” alone is not yet a sufficiently precise semantic assumption for expressivity claims.
>
> > My main concern is that the paper's positioning is stronger than what is actually established...
>
> We agree that the current positioning does not separate the contributions sharply enough. Our claim is not merely that finite precision matters in some broad informal sense, but that without fixing arithmetic semantics - evaluation order, rounding mode, overflow - the computational object being analyzed is not fully determined, and expressivity claims can be underspecified or misleading. Prior bounded-precision work does not characterize arithmetic at this level.
>
> All of the main components of our treatment are, to our knowledge, novel: treating arithmetic semantics explicitly as part of the model semantics for formal expressivity; the resulting semigroup-based formalization; the realized wreath-product perspective, which distinguishes the full abstract cascade from the subsemigroup induced by a fixed architecture, wiring, and arithmetic semantics; and the ensuing expressivity analysis for the diagonal SSM case study under our model, which both recovers and in some cases overturns prior analyses of the same architecture.
>
> > From a mathematical perspective, the framework is much stronger on upper bounds than on actual characterization…
>
> The meta-paradigm outlined in Section 6 explicitly has two parts: derive a structural upper bound via the realized wreath product, then obtain lower bounds by exhibiting concrete subsemigroups, groups, and wiring maps inside the model. We illustrated this second step in Section 7.4 with an explicit constructive separation. We agree, however, that the current presentation emphasizes the containment side more than the realizability side and will address this in revision.
>
> > I also think the paper moves too quickly from arithmetic sensitivity to broad conclusions about practical expressivity...
>
> Our expressivity claims apply to realized models with fixed hyperparameters: fixed state dimension, evaluation order, arithmetic semantics, precision, and no auxiliary memory growth. We do not claim this extends to indexed families in which width, depth, or precision grows with the input. We will revise the paper to make that distinction explicit.
>
> > Finally, the scope of the concrete results is narrow relative to the ambition of the paper...
>
> The diagonal SSM results are a case study illustrating how the framework simplifies and sharpens expressivity arguments, not a standalone collection of separation results. The main contribution is the algebraic framework itself.
>
> As a concrete illustration of why arithmetic semantics matter beyond abstract containment: as noted in our response to reviewer BfBZ, under saturation semantics, even a nominally linear recurrent update can acquire effective ReLU-like behavior, moving the same symbolic recurrence into a different expressivity regime.
>
> > Add a detailed comparison with NNTM / neural stack work... Also have comparison against this recent work...
>
> We will add a discussion of both lines of work. Our framework targets arithmetic characterization of recurrent models without augmented memory, making it complementary rather than competing. The work by Svete et al. extends the Minsky construction to Elman RNNs under continuity assumptions; we will cite it as related work.
>
>
> We will revise the paper throughout to sharpen the positioning, make the lower-bound methodology more explicit, and clarify the scope of our claims as described above.

---

> > ### Author Rebuttal · Reviewer_KwQy · 2026-04-03
> >
> > Assuming authors address my concerns, distinguish their work compared to other methods based on precision and time such as NNTM which is claimed to be smallest NN to simulate UTM would be interesting. I will increase my rating.

---

### Decision · Program_Chairs · 2026-04-30

**Decision:**

Accept (regular)

**Comment:**

After considering the reviews, rebuttal, and reviewer discussion, I recommend to accept this submission. The paper raises an important and timely issue in the theory of recurrent language models: expressivity claims depends on arithmetic semantics and evaluation order, and may therefore be underspecified or misleading. This methodological contribution is significant and the proposed algebraic framework offers a novel way to analyze these questions.

The discussion was helpful in clarifying both the paper’s strengths and its main weakness. The two key concerns were that the case study is narrow and the paper can be difficult to read. The reviewer discussion overall revealed these are limitations of scope and presentation but not fatal flaws in the contribution itself. In particular, the positive reviewers consistently emphasized that the framework is interesting, original, and potentially valuable beyond diagonal SSMs.

Overall, while the paper would benefit from a clearer presentation and stronger accessibility for a general ICML audience, it introduces a useful theoretical perspective, identifies an important potential source of confusion in the literature, and provides a framework that others are likely to build on. I encourage the authors to use the camera-ready version to improve readability, sharpen the distinction between established and new results, and make the broader applicability of the framework as concrete as possible.